# An orthogonal transcription mutation system generating all transition mutations for accelerated protein evolution in vivo

Mingwei Shao [1], Zhongnan Zhang [1], Xiaofan Jin [1], Jun Ding [1] & Guo-Qiang Chen [1,2,3,4,5] ✉

Targeted mutagenesis systems are critical for protein evolution. Current deaminase-T7 RNA polymerase fusion systems enable gene-specific mutagenesis but remain limited to certain model organisms. Here, we develop an orthogonal transcription mutation system for in vivo hypermutation in both non-model organism *Halomonas bluephagenesis* and *E. coli*, achieving >1,500,000-fold increased mutation rates. By fusing deaminases with three phage RNA polymerases, this system uniformly introduces C:G to T:A and A:T to G:C mutations across target genes. The system demonstrates high specificity, minimal off-target effects, and high orthogonality between phage polymerases. We apply this system to rapidly evolve fluorescent proteins, chromoproteins, cytoskeletal proteins, cell division-related proteins, global sigma factor, and the LysE exporter within a single day of the mutagenesis process. Overall, the orthogonal transcription mutation system is a modular and versatile platform that accelerates protein evolution in the shortest period reported so far.

Directed evolution is an effective approach to achieve accelerated natural evolution[1]. Biomolecules with enhanced functions are obtained by mutation and selection under laboratory conditions[2]. Traditional directed evolution methods, such as error-prone PCR, involve constructing variant libraries of target genes in vitro, followed by screening in host cells[3]. However, this approach is time-consuming and labor-intensive, and the size of the variant library is constrained by low transformation efficiency. To address these issues, some in vivo targeted mutation tools have been developed to enable continuous mutation, expression, and screening processes, thus increasing the mutagenesis efficiency and expanding the diversity of variant libraries. For example, in vivo targeted mutation approaches based on CRISPR-Cas technology take advantage of the sequence recognition specificity of sgRNA to generate mutations in specific regions, such as EvolvR[4], CRISPR-X[5], base editor[6], and prime editor[7]. However, these methods have a relatively narrow editing

window and often require the design of many gRNAs to cover the target gene. Orthogonal error-prone DNA polymerase-based methods such as OrthoRep[8] in *Saccharomyces cerevisiae*, BacORep[9] in *Bacillus thuringiensis*, and EcORep[10] in *Escherichia coli*, as well as virus replication-based methods such as PACE (phage-assisted continuous evolution)[11] and VEGAS (viral evolution of genetically actuating sequences)[12] have been successfully established. However, these approaches mutate the entire linear plasmid or viral genome, and as a result, they cannot achieve targeted mutation of specific genes or regions.

For efficient mutagenesis of specific regions, in vivo targeted mutation methods based on deaminase-fused T7 phage RNA polymerase (T7RNAP) were successfully established, such as rAPOBEC1 cytosine deaminase-based MutaT7[C→T][13], PmCDA1 cytosine deaminase-based eMutaT7[C→T][14], TadA8e adenine deaminase-based eMutaT7[A→G][15], eMutaT7[transition][15], and MutaT7[GDE][16] for all transition mutations. This

[1]School of Life Sciences, Tsinghua University, Beijing, China. [2]Department of Chemical Engineering, Tsinghua University, Beijing, China. [3]Center for Synthetic and Systems Biology, Tsinghua University, Beijing, China. [4]Tsinghua-Peking Center for Life Sciences, Beijing, China. [5]State Key Laboratory of Green Biomanufacturing, Beijing, China. ✉e-mail: chengq@mail.tsinghua.edu.cn

strategy has also been applied in mammals (TRACE)[17], yeast (TRIDENT)[18], and plants[19]. One of their applications was to evolve cis-aconitate decarboxylase successfully[20]. However, the current deaminase fused phage RNA polymerase strategy is limited to the T7RNAP, and it remains unclear whether this strategy can be extended to other phage RNAPs. In addition, for some non-model organisms such as *Halomonas* and *Pseudomonas*, the T7RNAP-based expression system is inefficient, limiting the application of in vivo targeted mutation approaches based on T7RNAP[21,22]. For example, in *Halomonas bluephagenesis*, T7RNAP failed to transcribe genes downstream of the $P_{T7}$ promoter[21], and the T7RNAP-based targeted mutation method established in *Pseudomonas putida* had a low mutation efficiency, which was only about 566-fold higher than the control[23].

To address the low transcription efficiency of the T7RNAP expression system in non-model organisms, broad-host expression systems based on MmP1, K1F, and VP4 phage RNAPs have been successfully developed in *H. bluephagenesis* and *Pseudomonas entomophila*[21]. These expression systems can also be applied to express heterologous genes efficiently in *E. coli*[21], *Comamonas testosteroni*[24], and *P. putida*[25]. *H. bluephagenesis* is a halophile isolated from Aydingol Lake in China that can grow in high-salt and alkaline conditions[26,27]. As a chassis for the Next Generation Industrial Biotechnology (NGIB), recombinant *H. bluephagenesis* allows the production of various types of polyhydroxyalkanoates (PHA)[28], small molecular products, and proteins[29] under open and non-sterile conditions, thus reducing production costs and process complexity. However, there are currently no targeted evolutionary tools available for non-model *H. bluephagenesis*.

Here, we construct the orthogonal transcription mutators (OTM) based on three phage RNA polymerases: MmP1, K1F, and VP4. These mutators can uniformly induce C:G to T:A and A:T to G:C transition mutations all over the target genes or pathways and exhibit high orthogonality in both *H. bluephagenesis* and *E. coli*. This system is studied to induce mutations in fluorescent proteins, chromoproteins, cytoskeletal proteins, cell division-related proteins, the $\sigma^{70}$ factor, and the arginine exporter, with consistently high efficiency observed across all applications.

## Results

### Construction of single type mutators for C:G to T:A mutations using PmCDA1 variant-MmP1 RNAP fusions

The C:G to T:A mutator plasmids were constructed based on the MmP1 phage RNAP[30] and PmCDA1 variants in a high copy-number plasmid (pSEVA241). The cytosine deaminase PmCDA1[31,32], PmCDA1-UGI fusion, and evoPmCDA1-UGI fusion[33] were fused to the N-terminus of MmP1 RNAP using an XTEN linker, respectively. They are designated as pMT1-MmP1, pMT2-MmP1, and pMT2.1-MmP1 (Fig. 1a). The uracil glycosylase inhibitor (UGI) prevents the removal of uracil produced by cytosine deamination via inhibiting the function of uracil DNA glycosylase (UDG)[34,35]. Blank vector pSEVA241 (p241) and the plasmid expressing MmP1 RNAP (pMT0-MmP1) were used as the control group. These PmCDA1 variant-phage RNAP fusions and MmP1 RNAP were driven by the IPTG-inducible tac promoter ($P_{Tac}$).

To evaluate the transcriptional activity of the single mutators, a target plasmid was constructed expressing sfGFP under the MmP1 promoter ($P_{MmP1}$) for induction studies and flow cytometry analysis (Fig. 2a and Supplementary Fig. 1a). The fluorescence intensity of the pMT1-MmP1, pMT2-MmP1 and pMT2.1-MmP1 constructs was comparable, approximately half that of the positive control pMT0-MmP1 (Fig. 2b and Supplementary Fig. 1). Compared to pMT0-MmP1, a notable portion of cells in pMT1-MmP1, pMT2-MmP1 and pMT2.1-MmP1 groups exhibited loss of fluorescence, probably due to the inactivation of sfGFP by the single type mutators (Fig. 2b and Supplementary Fig. 1). Therefore, these fusion proteins maintained normal transcriptional activity in *H. bluephagenesis*.

To assess the C:G to T:A type mutation rates in *H. bluephagenesis*, a mutation-recovery method was designed based on the erythromycin resistance gene (*ermC*). The *ermC* gene was inactivated via a missense mutation on its active site Y104. Briefly, the Y104S mutation demonstrated a complete inactivation, while the Y104F mutation retained erythromycin resistance[36] (Supplementary Fig. 2). The ErmC Y104S mutant was cloned and placed under the $P_{MmP1}$ promoter and a constitutive $P_{porin42}$ promoter in pSEVA321 as the target plasmid. It was observed that a correction of C to T at position 104 restored erythromycin resistance, changing the serine (S104, TCT) to phenylalanine (F104, TTT) (Fig. 2a). The on-target mutation frequency was $3.1 \times 10^{-7}$ for the control pMT0-MmP1, $1.9 \times 10^{-5}$ for pMT1-MmP1, $2.5 \times 10^{-2}$ for pMT2-MmP1, and $7.4 \times 10^{-4}$ for pMT2.1-MmP1, respectively. The pMT2-MmP1 (PmCDA1-UGI-MmP1) exhibited the highest mutation frequency, exceeding that of the control by over 80,000-fold (Fig. 2c). Compared to pMT1-MmP1 expressing PmCDA1-MmP1, the inclusion of UGI significantly enhanced mutation activity (over 1000-fold increase in mutation frequency) (Fig. 2c). Additionally, the maximum likelihood method was employed to recalculate the mutation rate, and the mutation rates of pMT2-MmP1 was $2.9 \times 10^{-5}$ substitutions per base (s.p.b.). Furthermore, evoPmCDA1 was demonstrated to be more efficient in *E. coli*, yet it failed to achieve a higher mutation activity in *H. bluephagenesis*[37] (Fig. 2c).

The rifampicin-resistant mutation frequency was employed to evaluate the off-target rate within the genome. Interestingly, the off-target rates of pMT1-MmP1 and pMT2-MmP1 increased only 14-fold and 5-fold compared to the control pMT0-MmP1, indicating their high specificity (Fig. 2d). In contrast, the off-target phenomenon was more pronounced in pMT2.1-MmP1, exhibiting a 154-fold increase (Fig. 2d). Effects on cell viability were evaluated by measuring colony-forming units (CFU/mL): pMT1-MmP1 and pMT2-MmP1 had negative impact on cell growth ($2.7 \times 10^8$ and $9.3 \times 10^7$ CFU/mL) compared to the control pMT0-MmP1 ($1.1 \times 10^9$ CFU/mL) (Fig. 2e). While pMT2.1-MmP1 showed no impact on cell viability ($2.4 \times 10^9$ CFU/mL) (Fig. 2e).

The effect of IPTG inducer concentration was investigated for the pMT2-MmP1 mutator exhibiting the highest on-target mutation frequency. Results indicated that the IPTG inducer concentration regulated the on-target and off-target mutation frequencies and cell viability. With increasing inducer concentration, the on-target mutation frequency was significantly increased to over 20,000-fold (Fig. 2f), while the off-target rate increased only 7-fold (Fig. 2g). Interestingly, high IPTG concentration dramatically reduced the cell count, decreasing from $5.9 \times 10^9$ CFU/mL without inducer to $5.9 \times 10^7$ CFU/mL with 200 mg/L IPTG (Fig. 2h). These results indicated that the on-target rates and cell viability were tunable by regulating inducer concentration.

### Optimization and expansion of C:G to T:A type mutators

The C-terminal of PmCDA1 was fused with UGI in the pMT2-MmP1 mutator to increase mutation activity. To evaluate the effect of UGI expression on mutation activity, the *ugi* gene was expressed independently with three different strengths of ribosome binding sites (RBSs) (Fig. 3a)[38]. The independent expression of UGI further increased the on-target mutation frequency, showing a 6- to 12-fold increase compared to pMT2-MmP1, while the off-target rate and cell number revealed no significant change (Fig. 3b and Supplementary Fig. 3). The pMT2-MmP1-W exhibited the highest mutation rate of $3.9 \times 10^{-4}$ s.p.b. (mutation frequency: $4.6 \times 10^{-1}$), which is more than 1,500,000 times greater than that of the control ($2.6 \times 10^{-10}$ s.p.b.).

The C:G to T:A type mutators were further constructed based on K1F and VP4 phage RNAPs (Fig. 3c). Both pMT2-K1F and pMT2-VP4 demonstrated high mutation rates of $9.6 \times 10^{-6}$ s.p.b. (mutation frequency: $1.4 \times 10^{-2}$) and $7.8 \times 10^{-6}$ s.p.b. (mutation frequency: $6.4 \times 10^{-3}$), respectively (Fig. 3d). The off-target frequencies of pMT2-K1F and pMT2-VP4 were observed with only weak increases of 3-fold and 12-fold, respectively (Fig. 3e). Notably, the cell numbers were comparable

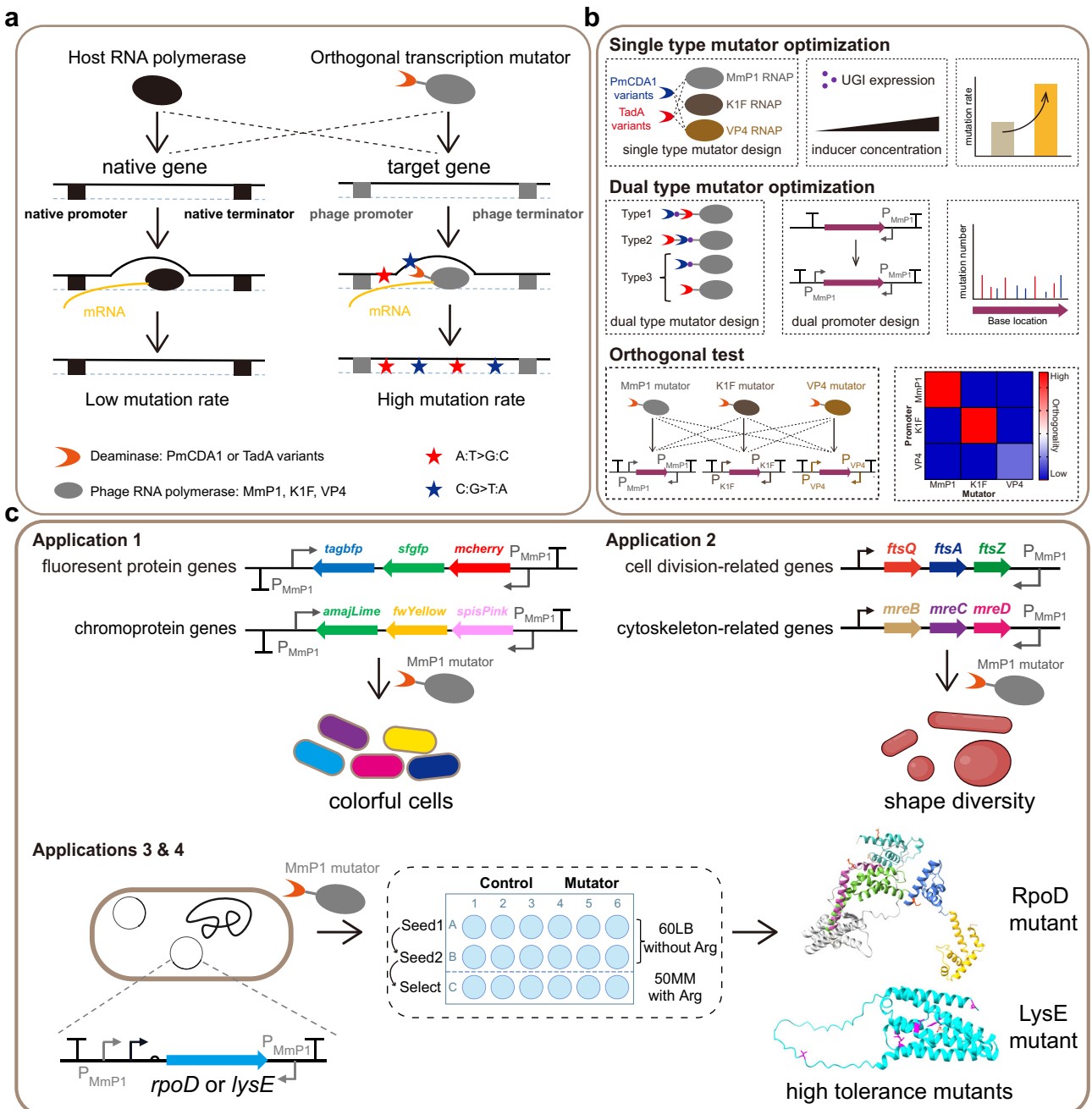

**Fig. 1 | Design, development, optimization, and application of the orthogonal transcription mutation system. a** Design and mechanism of the orthogonal transcription mutation system. The host RNAP recognizes the endogenous promoter and transcribes to produce mRNA with a low background mutation efficiency. Orthogonal transcription mutators are based on the fusion of deaminase PmCDA1 and TadA variants with phage RNAPs. The phage RNAP domain specifically recognizes the corresponding phage promoter and opens the double-stranded target gene. The deaminase domain generates random mutations on single-stranded DNA with a high mutation efficiency. **b** Systematic optimization of the orthogonal transcription mutation system. Optimization of single type mutators (C:G to T:A or A:T to G:C type mutators) by combining PmCDA1 and TadA variants with different phage RNAPs, regulating the expression of UGI, and controlling the inducer concentration to increase the mutation rate. Three dual type mutators were designed and tested to generate C:G to T:A and A:T to G:C mutations. Then, a dual promoter strategy was implemented to distribute mutations across the target gene evenly. Finally, orthogonality tests among three phage RNAP-based mutators demonstrated high specificity. **c** Applications of the orthogonal transcription mutation system. Orthogonal transcription mutators were used to mutate fluorescent proteins and chromoproteins to produce colorful cells, mutate cytoskeleton and cell division-related proteins to achieve morphological diversity, and mutate sigma 70 factor RpoD and LysE exporter to achieve higher tolerance to L-arginine.

between the mutator groups (pMT2-K1F, pMT2-VP4) and the controls (pMT0-K1F and pMT0-VP4), indicating that the utilization of different phage RNAPs can regulate their impact on cell viability (Fig. 3f). Therefore, C:G to T:A type mutators with high mutation activity, high specificity, and low cytotoxicity were successfully obtained after optimization.

## Construction of single type mutators for A:T to G:C mutations based on TadA variant-phage RNAP fusions

Three adenine deaminase TadA variants, namely, TadA7.10[39], TadA8e[40], and TadA9[41], were selected to construct A:T to G:C type mutators. The three deaminases were fused to the N-terminus of MmP1 RNAP through an XTEN linker, resulting in mutators termed

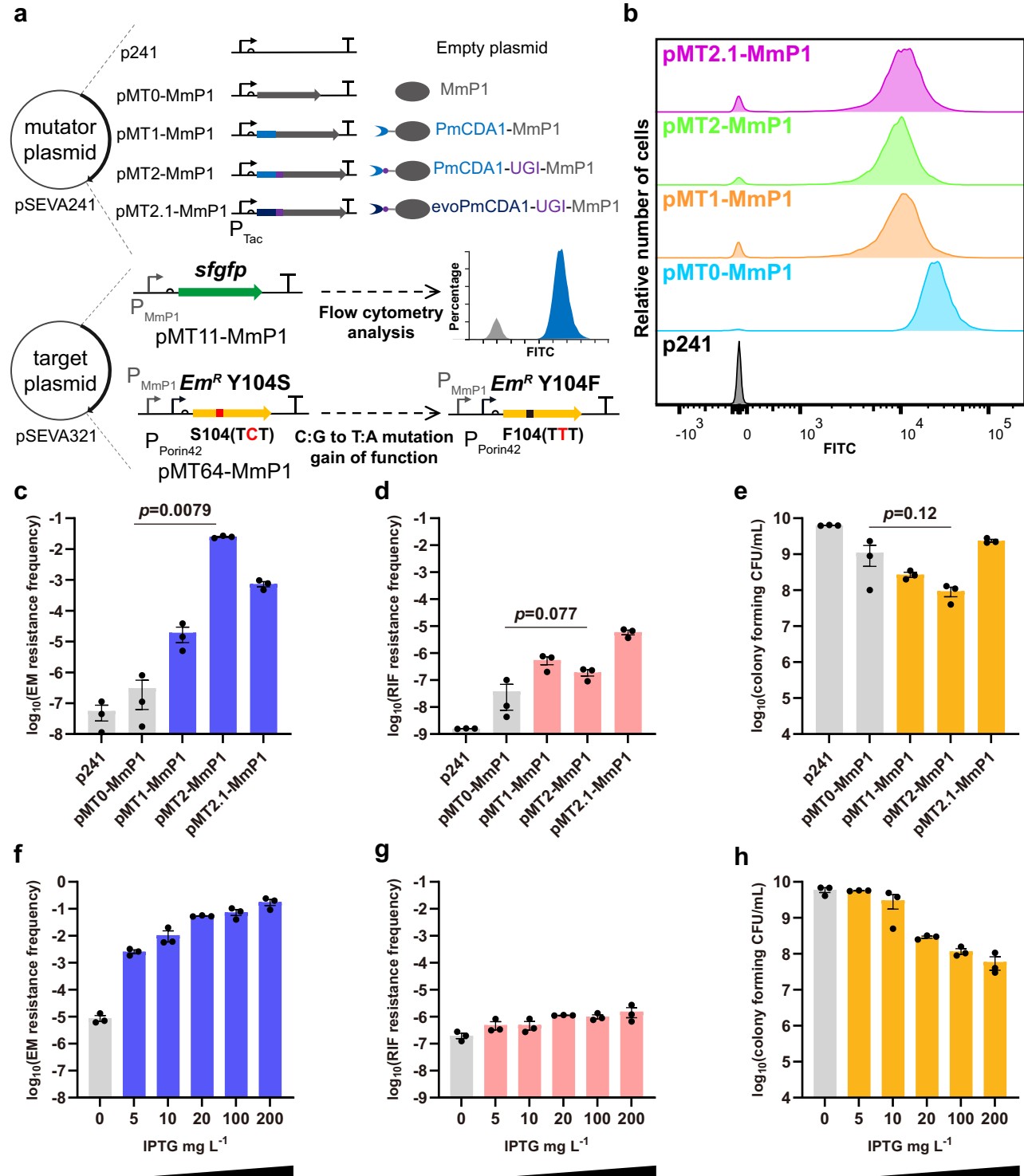

**Fig. 2 | Design and characterization of C:G to T:A type single mutators.**
**a** Construction of MmP1-based C:G to T:A type mutators by fusing cytosine dea-
minases to MmP1 RNAP via the XTEN linker. The sfGFP was used as the reporter to
determine the transcriptional activity of C:G to T:A type single mutators.
Erythromycin-resistant mutant ErmC Y104S was used to assess on-target mutation
rate, as C:G to T:A correction at position 104 restored the antibiotic function (S104
TCT to F104 TTT). **b** Transcriptional activity of C:G to T:A type single mutators
tested using flow cytometry after 12 h with 200 mg/L IPTG. **c**–**e** Analysis of the on-
target mutation rate (erythromycin resistance frequency, EM resistance

frequency), off-target rate (rifampicin resistance frequency, RIF resistance fre-
quency), and cell viability (colony forming CFU/mL) of C:G to T:A type single
mutators. **f**–**h** Analysis of the on-target mutation rate, off-target rate, and cell via-
bility of pMT2-MmP1 (PmCDA1-UGI-MmP1) at different inducer concentrations.
Data are presented as mean values (bars), standard errors (error bars), and indivi-
dual values (black dots). $n = 3$, which represents three independent replicates of the
experiment. Statistical analyses were conducted using two-tailed Student's t-tests.
A $p$-value < 0.05 was considered significant. Source data are provided as a Source
Data file.

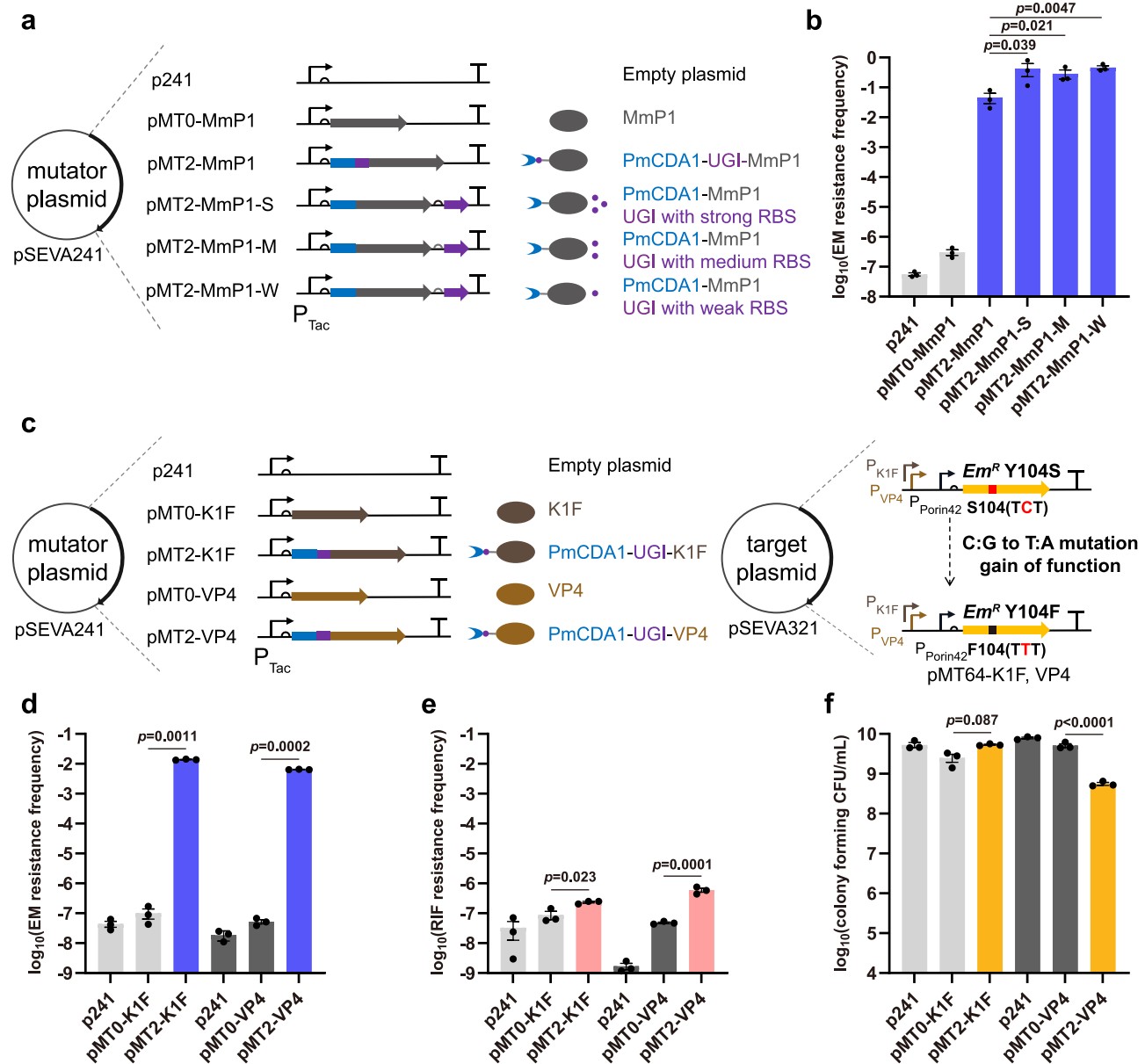

**Fig. 3 | Optimization and expansion of C:G to T:A type single mutators.**
**a** Optimization of the PmCDA1 mutator by independently expressing UGI with strong, medium, and weak RBSs. **b** Analysis of on-target mutation rates of PmCDA1 mutators with three different strengths of RBSs. **c** Expansion of C:G to T:A type single mutators via the fusion of PmCDA1-UGI to K1F and VP4 RNAPs, respectively. To assess the on-target mutation rate, Erythromycin resistance mutant ErmC Y104S was used under the control of $P_{K1F}$ and $P_{VP4}$ promoters, respectively. **d**–**f** Analysis of

the on-target mutation rate, off-target rate, and cell viability of K1F and VP4-based C:G to T:A type single mutators. Data are presented as mean values (bars), standard errors (error bars), and individual values (black dots). $n = 3$, which represents three independent replicates of the experiment. For statistical analyses, two-tailed Student's t-tests were applied. A $p$-value < 0.05 was considered significant. Source data are provided as a Source Data file.

pMT3.1-MmP1, pMT3-MmP1, and pMT3.4-MmP1, respectively (Fig. 4a). Flow cytometry was employed to evaluate the transcriptional activities of the three mutators. The blank vector pSEVA241 (p241) and the plasmid expressing MmP1 RNAP (pMT0-MmP1) served as negative and positive controls, respectively. Fluorescence intensities of pMT3-MmP1 and pMT3.4-MmP1 were observed to be comparable, approximately one-tenth of that of the positive control. In contrast, pMT3.1-MmP1 exhibited a much lower fluorescence intensity, suggesting weak transcriptional activity of TadA7.10-MmP1 mutator (Fig. 4b and Supplementary Fig. 4).

To assess the on-target mutation rates of A:T to G:C type mutators, ErmC Q10* mutant was cloned in the target plasmid (Fig. 4a and Supplementary Fig. 5). The A to G mutation at position 10 on the

complementary strand converts TAA (premature termination codon) to CAA (glutamine, Q), thereby restoring erythromycin resistance activity. The pMT3-MmP1 and pMT3.4-MmP1 were found to exhibit similar on-target mutation frequencies of $3.8 \times 10^{-3}$ and $8.6 \times 10^{-4}$, off-target mutation frequencies of $1.5 \times 10^{-6}$ and $1.2 \times 10^{-6}$, and cell number of $1.8 \times 10^{7}$ and $4.3 \times 10^{7}$ CFU/mL, respectively (Fig. 4c–e). The mutation rate of pMT3-MmP1 was $9.0 \times 10^{-6}$ s.p.b., which was about 67,000 times higher than that of the control. The elevated off-target rate and cell burden could be attributed to the high mutation activity of TadA8e and TadA9[40,41]. Additionally, the on-target mutation frequency of pMT3.1-MmP1 was relatively low at $1.8 \times 10^{-5}$ (Fig. 4c). However, pMT3.1-MmP1 presented a low off-target rate with only a 3-fold increase (Fig. 4d) and had a comparable cell count of $7.6 \times 10^{9}$ CFU/mL

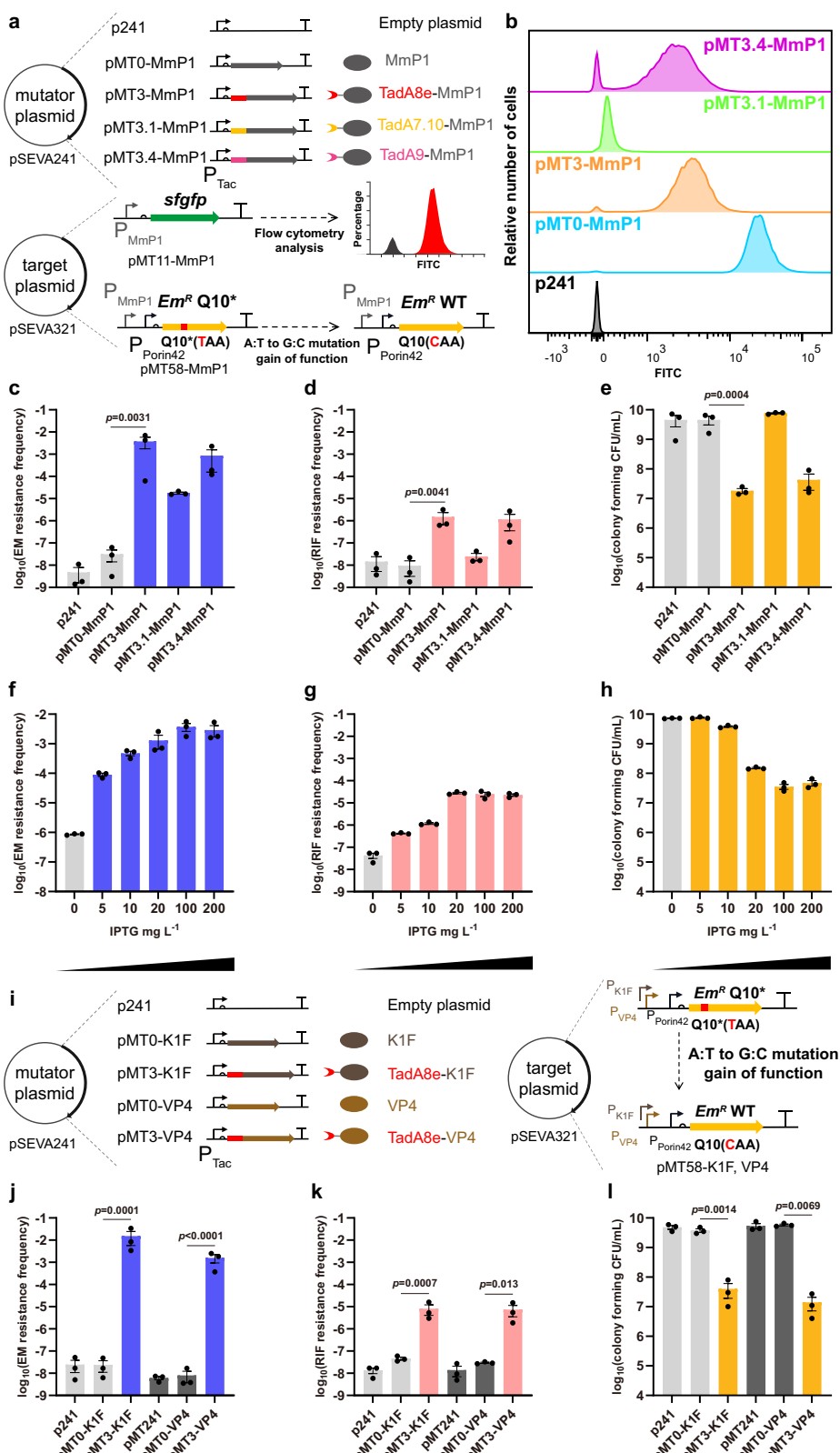

(Fig. 4e). These findings suggested that the combination of different adenine deaminases can regulate the on- and off-target mutation frequencies together with the cell viability.

The effect of inducer concentration on the pMT3-MmP1 mutator was subsequently investigated. With IPTG concentration increased from 0 to 200 mg/L, the on-target mutation frequency raised more than 3,300-fold, from $8.6 \times 10^{-7}$ to $2.9 \times 10^{-3}$ (Fig. 4f), the off-target mutation frequency increased 530-fold, from $4.3 \times 10^{-8}$ to $2.3 \times 10^{-5}$ (Fig. 4g), and the cell count decreased 153-fold, from $7.2 \times 10^9$ to $4.7 \times 10^7$ CFU/mL (Fig. 4h). The ratio of on-target mutation frequency to off-target mutation frequency reached its peak at 404-fold in the presence of 10 mg/L IPTG, indicating that an optimal on-target to off-target ratio can be achieved by adjusting the IPTG concentration. Additionally, varying induction times were studied but showed no

**Fig. 4 | Design, characterization, and expansion of A:T to G:C type single mutators. a** Construction of MmP1-based A:T to G:C type single mutators by fusing adenine deaminases to MmP1 RNAP using the XTEN linker. The sfGFP was used as the reporter to determine the transcriptional activity of A:T to G:C type single mutators. Erythromycin resistance mutant ErmC Q10* was used to assess the on-target mutation rate. **b** Transcriptional activity of A:T to G:C type single mutators tested by flow cytometry analysis. **c–e** Analysis of on-target mutation rate, off-target rate, and cell viability of A:T to G:C type single mutators. **f–h** Evaluation of on-target mutation rate, off-target rate, and cell viability of pMT3-MmP1 (TadA8e-MmP1) at different inducer concentrations. **i** Expansion of A:T to G:C type single mutators by fusing TadA8e to the N-terminus of K1F and VP4 RNAPs, respectively. To assess the on-target mutation rate, Erythromycin resistance mutant ErmC Q10* was used to assess the on-target rate under the control of $P_{K1F}$ and $P_{VP4}$ promoters, respectively. **j–l** Analysis of on-target mutation rate, off-target rate, and cell viability of K1F-and VP4-based A:T to G:C type single mutators. Data are presented as mean values (bars), standard errors (error bars), and individual values (black dots). $n = 3$, representing three independent replicates of the experiment. Statistical analyses were performed using two-tailed Student's t-tests. A $p$-value < 0.05 was considered significant. Source data are provided as a Source Data file.

significant difference in the off-target rate (Supplementary Fig. 6). When TadA8e was fused to the N-terminus of K1F RNAP and VP4 RNAP, the resulting mutators were named as pMT3-K1F and pMT3-VP4. The $P_{MmP1}$ promoter in the target plasmid was replaced with $P_{K1F}$ and $P_{VP4}$ promoters, respectively (Fig. 4i). The results of the mutation induction assay were revealed to be similar to those of pMT3-MmP1 (Fig. 4j–l). Both pMT3-K1F and pMT3-VP4 were shown with high on-target mutation frequencies, elevated off-target mutation frequencies, and low cell viability, likely due to the high mutation activity of TadA8e[40].

## Construction of dual type mutators for dual transition mutations

The type of mutation affects the diversity of the mutant library, which ultimately affects the outcome of directed evolution. More mutation types diversify the mutant library and increase the likelihood of obtaining the desired mutants. To achieve both C:G to T:A and A:T to G:C types of in vivo mutations, three strategies were adopted combining cytosine deaminase and adenine deaminase with MmP1 RNAP (Fig. 5a). PmCDA1-UGI and TadA8e were fused to MmP1 in different orders to form pMT23d-MmP1 and pMT32d-MmP1 dual type mutators. The pMT23opt-MmP1 dual type mutator was designed by expressing PmCDA1-UGI-MmP1 and TadA8e-MmP1 in the same plasmid. To avoid the loss of fragments due to homologous recombination, the sequence of MmP1 in TadA8e-MmP1 was codon-optimized based on the codon usage of *H. bluephagenesis* (Fig. 5a). ErmC Y104S and ErmC Q10* mutants were used to assess the C:G to T:A and A:T to G:C type mutation frequencies. The negative control pMT0-MmP1 and positive controls pMT2-MmP1 and pMT3-MmP1 were included in the subsequent mutation induction assays. Compared to the controls, the mutation frequencies for converting C:G to T:A and A:T to G:C types in pMT23d-MmP1 were elevated by $3.9 \times 10^3$-fold and $1.3 \times 10^5$-fold, suggesting a strong preference for A:T to G:C type mutations (Fig. 5b). Mutation frequencies of C:G to T:A and A:T to G:C types in pMT32d-MmP1 increased by $2.4 \times 10^5$-fold and $7.6 \times 10^4$-fold, respectively, indicating a slight preference for C:G to T:A type mutations (Fig. 5b). In contrast, the mutation frequencies for transforming C:G to T:A and A:T to G:C types in pMT23opt-MmP1 were comparable with increases of $2.8 \times 10^5$-fold and $5.0 \times 10^5$-fold, respectively (Fig. 5b). The mutation type preference observed in pMT23d-MmP1 and pMT32d-MmP1 demonstrated that the deaminase closer to MmP1 RNAP exhibited higher mutation activity, likely due to the increased chance of the deaminase acting on single-strand DNA during transcription.

To evaluate the number and distribution of mutations in the target gene generated by dual type mutators, a selection method based on the *sacB* reporter gene was established (Fig. 5c). The *sacB* gene encodes levansucrase, which converts sucrose into the toxic product levan, leading to cell death[42]. The reporter gene under the constitutive $P_{porin42}$ promoter was inserted into the genomic locus G7 of *H. bluephagenesis* TD01 (Supplementary Fig. 7). The $P_{MmP1}$ promoter was placed downstream of the *sacB* gene in reverse orientation to recruit dual type mutators. The on-target mutation frequencies of pMT23d-MmP1, pMT32d-MmP1, and pMT23opt-MmP1 were $6.6 \times 10^{-2}$, $1.2 \times 10^{-2}$, and $2.1 \times 10^{-1}$, respectively (Fig. 5d). The

pMT32d-MmP1 and pMT23opt-MmP1 exhibited low off-target rates, while pMT23d-MmP1 a significantly higher off-target rate (Supplementary Fig. 8a and b). This may be due to the high mutation activity of adenine deaminase TadA8e in pMT23d-MmP1, which was closer to MmP1 RNAP.

DNA sequencing of *sacB* mutants indicated that pMT32d-MmP1 and pMT23opt-MmP1 generated both C:G to T:A and A:T to G:C type mutations, whereas only A:T to G:C type mutations were found in *sacB* mutants from pMT23d-MmP1 (Fig. 5e), possibly due to the strong preference of pMT23d-MmP1 for A:T to G:C type mutations. No insertion or deletion mutations were detected in *sacB* mutants generated by these three dual type mutators, whereas a substantial number of transposon insertion mutations were observed in colonies derived from the control pMT0-MmP1 (Supplementary Fig. 9). Furthermore, the average number of mutations per colony was 1.4, 1.3, and 2.9 in pMT23d-MmP1, pMT32d-MmP1, and pMT23opt-MmP1, respectively, demonstrating that pMT23opt-MmP1 possesses higher mutagenesis activity (Fig. 5f).

The distribution of mutations on the *sacB* gene was also evaluated. Interestingly, three dual type mutators were more likely to generate mutations in regions close to the $P_{MmP1}$ promoter than that in regions far from the promoter (Fig. 5g). This phenomenon may be attributed to the decreasing transcriptional activity of MmP1 RNAP during transcription. Subsequently, a dual promoter strategy was employed by positioning two $P_{MmP1}$ promoters both upstream and downstream of the *sacB* gene (Fig. 5h). The dual promoter strategy was observed to further enhance the on-target mutation rate and increase the average number of mutations per colony (Fig. 5i–k). Additionally, the mutations were found evenly distributed across the *sacB* gene (Fig. 5l).

To further investigate the mutagenesis characteristics of three orthogonal transcription mutators, next-generation sequencing (NGS) was employed to analyze the 3.5 kb DNA fragments containing the *sacB* gene after mutation induction assay. To investigate terminator effect on mutation distribution, four terminators were placed downstream of *sacB*, while one terminator was placed upstream (Fig. 6a). NGS analysis revealed that pMT23opt-MmP1 produced balanced substitution frequencies: 0.68% for A to G, 0.51% for T to C, 0.22% for C to T, and 0.19% for G to A (Fig. 6b). However, pMT23d-MmP1 and pMT32d-MmP1 exhibited a strong preference for A:T to G:C and C:G-T:A mutations, respectively, which was consistent with previous mutation rate assays (Figs. 5b, 6b). The 3.5 kb DNA fragment was divided into three regions: upstream (1–961), target (962–2985), and downstream (2986–3585). Compared to the control pMT0-MmP1, pMT23opt-MmP1 exhibited a significantly higher transition substitution frequency in the target region (0.22% vs. 0.06%), with a comparable frequency (0.06% vs. 0.05%) in the downstream region and an increased frequency in the upstream region (0.09% vs. 0.05%) (Fig. 6c, f and Supplementary Fig. 10). The elevated upstream mutation frequency is associated with insufficient transcription termination, suggesting that four terminators are sufficient to effectively terminate the transcription process. pMT23d-MmP1 and pMT32d-MmP1 exhibited similar mutation distributions, but pMT23d-MmP1 demonstrated a higher transition substitution

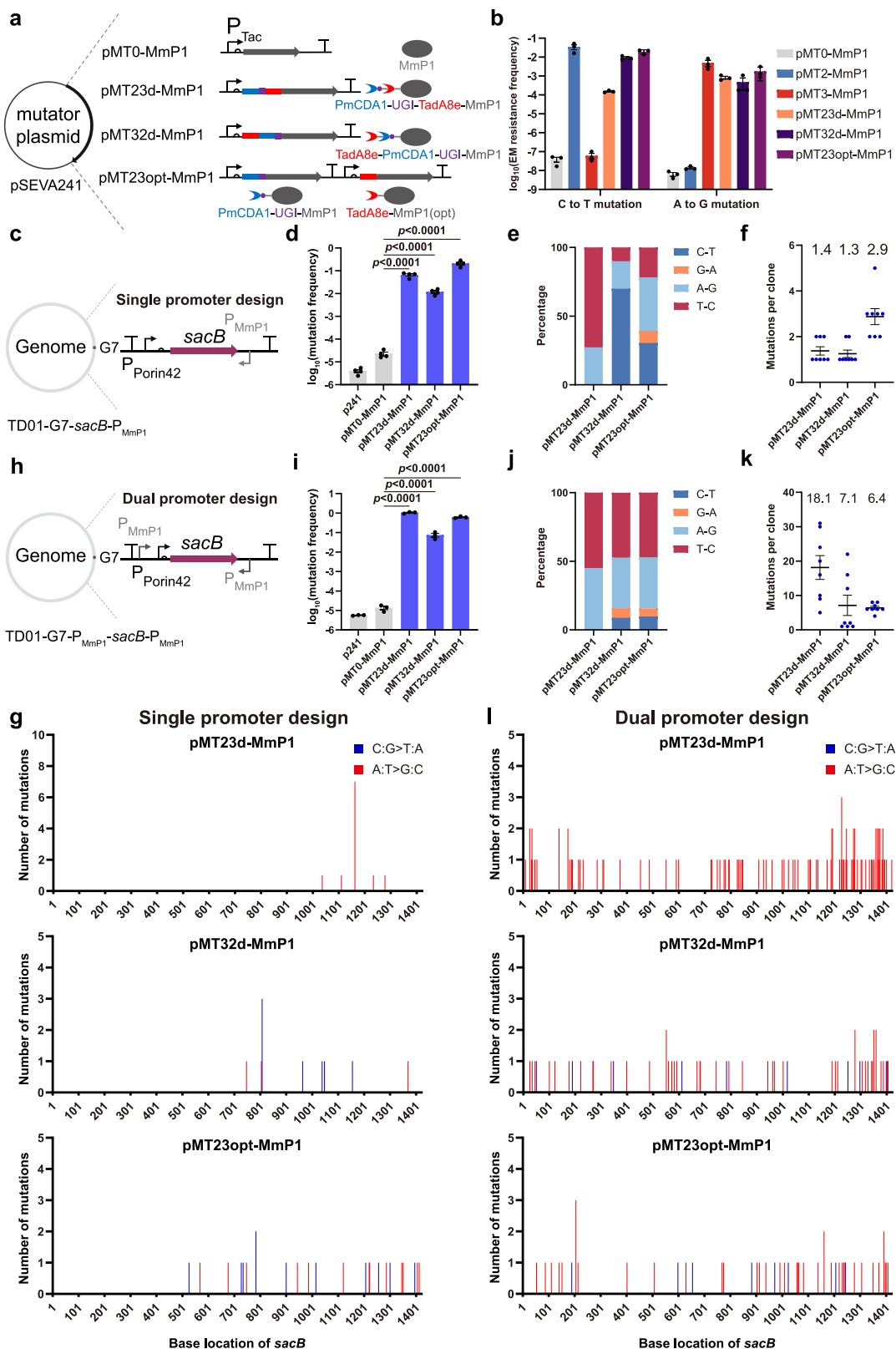

frequency in the target region (1.34%), whereas pMT32d-MmP1 had a lower mutation frequency (0.07%), consistent with previous sequencing results (Fig. 6d, e and Supplementary Fig. 10). In summary, the dual type mutators were capable of generating both C:G to T:A and A:T to G:C type mutations uniformly across the target gene when two $P_{MmP1}$ promoters were positioned both upstream and downstream of the gene.

## Expansion of dual type mutators and orthogonality studies

To test the compatibility of dual type mutator strategies, PmCDA1-UGI and TadA8e were extended to K1F and VP4 RNAPs (Fig. 6g and i). The dual type mutators based on K1F and VP4 produced both C:G to T:A and A:T to G:C type mutations (Fig. 6h and j). Further characterization using the *sacB* gene indicated that three dual type mutators based on K1F and VP4 generated a substantial number of evenly distributed

**Fig. 5 | Design and characterization of C:G to T:A and A:T to G:C dual type mutators. a** Design of three types of dual mutators by fusing PmCDA1-UGI and TadA8e to MmP1 RNAP in different orders or co-expressing PmCDA1-UGI-MmP1 and TadA8e-MmP1 (opt) within the same plasmid. The term "opt" refers to codon optimization. **b** The mutation frequencies of C:G to T:A and A:T to G:C types for three dual mutators were assessed by erythromycin resistance frequency ($n = 3$ independent experiments). **c** Construction of a *sacB*-based single $P_{MmP1}$ promoter selection method in the genomic locus G7 of *H. bluephagenesis* TD01 (TD01-G7-*sacB*-$P_{MmP1}$). The expression of the *sacB* gene was placed under the $P_{Porin42}$ constitutive promoter, with a $P_{MmP1}$ promoter positioned downstream in the reverse direction. **d–f** Mutation frequency ($n = 4$ independent experiments), mutation types, and the average number of mutations ($n = 8$ independent experiments) in *sacB* were determined using the selection method described in (**c**). **g** Distribution of mutations on *sacB* gene using selection method described in (**c**) ($n = 8$ independent experiments). **h** Construction a *sacB*-based dual pMmP1 promoters selection method at the genomic locus G7 of *H. bluephagenesis* TD01 (TD01-G7-$P_{MmP1}$-*sacB*-$P_{MmP1}$). The expression of the *sacB* gene was placed under the $P_{Porin42}$ constitutive promoter, with two PMmP1 promoters placed both upstream and downstream of the *sacB* gene. **i–k** Mutation frequency ($n = 3$ independent experiments), mutation type, and average number of mutations ($n = 8$ independent experiments) in *sacB* were determined using the selection method described in (**h**). **l** Distribution of mutations on *sacB* gene using selection method described in (**h**) ($n = 8$ independent experiments). Data are presented as mean values (bars), standard errors (error bars), and individual values (black or blue dots). Statistical analyses were conducted using two-tailed Student's t-tests. A *p*-value < 0.05 was considered significant. Source data are provided as a Source Data file.

mutations on *sacB* gene (Supplementary Figs. 11, 12). These findings suggested that dual type mutator strategies can be easily extended to other phage RNAPs.

Subsequently, an orthogonality study was designed to evaluate the interference among mutators based on three RNAPs (Fig. 6k). Three dual type mutators, namely, pMT23opt-MmP1, pMT23opt-K1F, and pMT23opt-VP4, along with three control plasmids expressing only RNAPs, were transferred into three reporter strains *H. bluephagenesis* TD01-G7-$P_{MmP1}$-*sacB*-$P_{MmP1}$, TD01-G7-$P_{K1F}$-*sacB*-$P_{K1F}$, and TD01-G7-$P_{VP4}$-*sacB*-$P_{VP4}$, for the mutation induction assays, respectively. The fold change in mutation frequencies relative to the control was utilized to evaluate the orthogonality among dual type mutators. When recognizing the corresponding promoters $P_{MmP1}$, $P_{K1F}$, and $P_{VP4}$, the mutation frequencies of pMT23opt-MmP1, pMT23opt-K1F, and pMT23opt-VP4 improved by 46,608, 46,470, and 8,388 times, respectively (Fig. 6l). When recognizing non-corresponding promoters, however, the mutation frequencies increased only 6- to 27-fold compared to the control (Fig. 6l), indicating that dual type mutators based on different RNAPs exhibited high specificity with minimal crosstalk.

Finally, these dual type mutators were applied to *E. coli*. The dual type mutators based on MmP1, K1F, and VP4 RNAPs demonstrated high mutation activity in *E. coli* MG1655, generating mutations on *sacB* uniformly with minimal impact on cell viability (Supplementary Figs. 13–15). The results of the orthogonality tests indicated that these dual type mutators based on different RNAPs exhibited strong specificity and orthogonality in *E. coli* (Supplementary Fig. 16). In conclusion, the dual type mutator strategy could be expanded to various phage RNAPs and applied to both the non-model organism *H. bluephagenesis* and model organism *E. coli*. Additionally, dual type mutators based on different phage RNAPs specifically recognized their corresponding promoters, demonstrating high orthogonality.

### Evolution of genes encoding fluorescent protein and chromoprotein for color variance

To investigate the efficiency of the mutators, genes encoding fluorescent proteins and chromoproteins were used as targets to induce diversity in coloration. Two target plasmids were constructed based on the pSEVA321 vector, containing three genes encoding fluorescent proteins mCheery, sfGFP, and TagBFP, together with three genes encoding chromoproteins amajLime, fwYellow, and spisPink, respectively. These two gene clusters were driven by a strong constitutive $P_{Porin141}$ promoter, with two $P_{MmP1}$ promoters placed upstream and downstream of the gene clusters, respectively. Subsequently, *E. coli* MG1655 strains harboring each target plasmid were transformed with either the pMT23opt-MmP1 mutator plasmid or the pMT0-MmP1 control plasmid for induction experiments and color variation observation (Fig. 7a).

Under confocal microscopy, it was observed that cells with pMT23opt-MmP1 produced single fluorescent colors like red, blue, and green, together with combinations that resulted in yellow, purple, and cyan variations, with notable changes in brightness. However, the

color variation was not obvious in the control group (Fig. 7b and Supplementary Fig. 17). In addition, colonies with color changes on the plates were selected for culture, followed by confocal microscopy observation, colony PCR, and DNA sequencing analysis. It was identified with multiple mutations on the genes encoding mCheery, sfGFP, and TagBFP (Fig. 7c and Supplementary Figs. 18 and 19). For example, mutant 1 had the Q184* mutation in sfGFP, leading to the loss of green fluorescence, while mutant 2 was revealed with three amino acid mutations (V132M, M160I, and R179K) in TagBFP, affecting the function of BFP (Supplementary Table 4). Targeted mutations on three chromoprotein genes also induced a wide range of color variations to form purple, orange, pink, and white colonies of *E. coli* MG1655 (Fig. 7d, e). DNA sequencing of clones with color changes revealed numerous mutations on these chromoprotein genes (Fig. 7f, Supplementary Fig. 20, and Supplementary Table 5). The target plasmid containing three fluorescent protein genes and the mutator plasmid were also introduced into *H. bluephagenesis* TD01. A spectrum of colors was observed under confocal microscopy by mutating these fluorescent protein genes in *H. bluephagenesis* (Supplementary Fig. 21). The dual type mutators demonstrated the ability to rapidly induce mutations in fluorescent and chromoproteins, generating a wide range of color variations.

### Evolution of cytoskeleton and cell division-related genes for changing cell morphology

Morphology engineering enlarges or elongates cell shapes, allowing more intracellular products to be accumulated[43]. These morphologically modified cells are also easier to isolate, thereby reducing downstream processing costs[44]. Since genes related to the cytoskeleton and cell division are essential, traditional approaches in morphology engineering are limited in changing the expression levels of the cytoskeleton or cell division-related genes[43,45]. Here, cell morphology was investigated for manipulation via targeted mutation and evolution.

Cytoskeleton-related genes *mreBCD* and cell division-related genes *ftsQAZ* were selected as target genes for mutation. $P_{MmP1}$ promoter was placed downstream of *mreBCD* and *ftsQAZ* gene clusters in *H. bluephagenesis* TD01 either individually or simultaneously. The resulting strains were named *H. bluephagenesis* TD01-*mreBCD*-$P_{MmP1}$, TD01-*ftsQAZ*-$P_{MmP1}$, and TD01-*mreBCD*& *ftsQAZ*-$P_{MmP1}$ (Fig. 8a). Subsequently, the dual type mutator pMT23opt-MmP1 and the control pMT0-MmP1 were transferred to the above strains for induction studies, followed by morphology observation under optical microscope and scanning electron microscope (SEM). As expected, some cells significantly elongated when the *ftsQAZ* gene cluster was mutated, with cell lengths found to be over 20 μm. However, no elongated cells were observed in the controls. Similarly, mutations in the *mreBCD* gene cluster generated enlarged cells with a change in cell morphology from rod-shaped to spherical ones. Interestingly, cells exhibited a wide variety of morphological changes, including long rods, spheres, and other irregular shapes

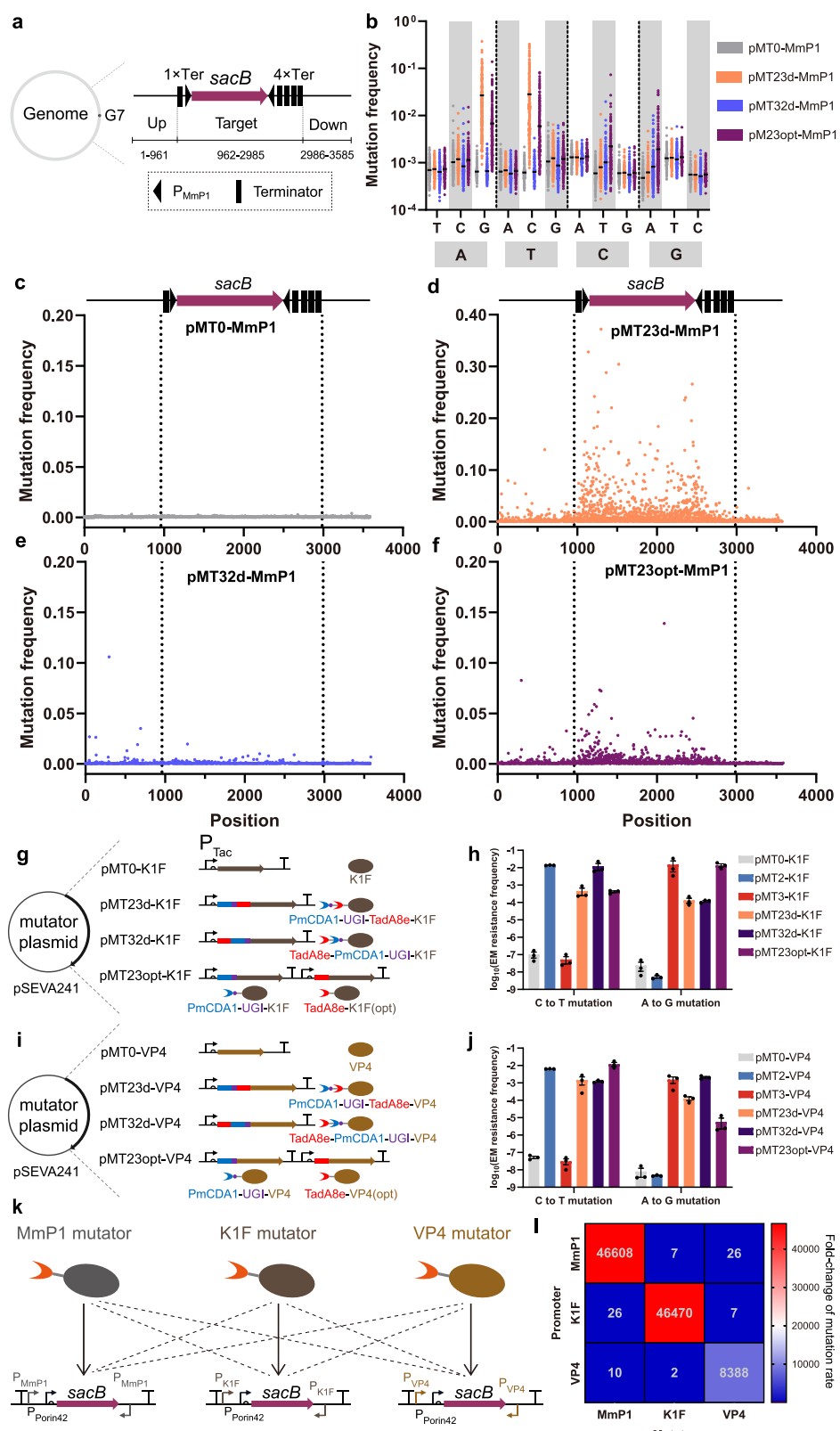

when both the *ftsQAZ* and *mreBCD* gene clusters were mutated (Fig. 8b and Supplementary Fig. 22). Subsequently, several mutants that exhibited morphological changes were selected for DNA sequencing, and mutations were found in the *ftsQAZ* and *mreBCD* gene clusters (Supplementary Figs. 23–25 and Supplementary Tables 6–9). The $P_{MmP1}$ promoter was also placed downstream of *mreBCD* and *ftsQAZ* gene clusters in *E. coli* MG1655, respectively,

referred to as MG1655-*mreBCD*-$P_{MmP1}$ and MG1655-*ftsQAZ*-$P_{MmP1}$. Cells were elongated and enlarged by mutating these two clusters, and they exhibited morphological changes similar to those of *H. bluephagenesis* (Supplementary Fig. 26). These results suggested that cell morphology can be modified easily by mutating *ftsQAZ* and *mreBCD* gene clusters using dual type mutators, resulting in the formations of enlarged and elongated cells.

**Fig. 6 | Next-generation sequencing (NGS) analysis of MmP1-based dual mutators, expansion of C:G to T:A and A:T to G:C type dual mutators, and orthogonality analysis. a** DNA fragment used for NGS. Ter represents terminator. **b** Transition substitution frequency of three orthogonal transcription mutators and the control within the target region. Mean value is represented by the short black line (ca. $10^5$ reads). **c–f** Location distribution of transition mutations in the 3.5 kb DNA fragments of pMT0-MmP1, pMT23d-MmP1, pMT32d-MmP1, and pMT23opt-MmP1. **g** Design of three types of dual mutators by fusing PmCDA1-UGI and TadA8e to K1F RNAP in different orders or co-expressing PmCDA1-UGI-K1F and TadA8e-K1F (opt) within the same plasmid. The term "opt" refers to codon optimization. **h** The C:G to T:A and A:T to G:C type mutation rates of three K1F-based dual mutators

tested by erythromycin resistance frequency ($n = 3$ independent experiments). **i** Design of three types of dual mutators by fusing PmCDA1-UGI and TadA8e to VP4 in different orders or coexpressing PmCDA1-UGI-VP4 and TadA8e-VP4 (opt) within the same plasmid. **j** The C:G to T:A and A:T to G:C type mutation rates of three VP4-based dual mutators tested by erythromycin resistance frequency ($n = 3$ independent experiments). **k** Orthogonality assessment scheme of dual type mutators based on three phage RNAPs. **l** Heat map profiling of fold-change in mutation rate by three dual type mutators, respectively. Data are presented as mean values (bars), standard errors (error bars), and individual values (black dots). Source data are provided as a Source Data file.

## Evolution of sigma 70 factor *rpoD* gene for higher L-Arginine tolerance

Global transcription machinery engineering (gTME) is a strategy that perturbs the global transcriptional regulatory network by modifying key transcription factors, such as sigma factors[46]. The main process of gTME involves creating mutant libraries of transcription factor genes and screening mutants with enhanced tolerance or increased yield[47]. In prokaryotes, the sigma 70 factor ($\sigma^{70}$) family, such as *rpoD*, is involved in regulating the expression of a wide variety of genes[48]. However, most studies use error-prone PCR to construct mutant libraries, which is a time-consuming and labor-intensive process. L-arginine (L-Arg) is a semi-essential amino acid widely utilized in the food additive, pharmaceutical, and medical industries[49]. However, high concentrations of L-Arg exhibit cytotoxic effects and inhibit cell growth[49]. Therefore, we utilized orthogonal transcription mutators to accelerate the mutation and selection process of *rpoD* in *H. bluephagenesis* (Fig. 8c).

The *rpoD* gene, along with native upstream promoter in *H. bluephagenesis*, was introduced into the endogenous plasmid of *H. bluephagenesis* as the target plasmid[50], with MmP1 promoter at both ends. The target plasmid, along with the dual mutator plasmid pMT23opt-MmP1 and the control plasmid pMT0-MmP1 was transferred into *H. bluephagenesis* TD01 strain, with its endogenous plasmid knocked out (TD01Δenp). These two strains were inoculated in 60LB medium without inducer as the first seed culture and were then diluted 100-fold in 60LB medium with 200 mg/L IPTG inducer and incubated at 37 °C for 24 h as the second seed culture to construct mutant libraries. Subsequently, the culture solution was diluted 100-fold in 50MM medium containing 6 g/L L-Arg to select mutants with enhanced L-Arg tolerance. Finally, the cell culture was spread on 50 MM solid plates containing 6 g/L L-Arg, and large colonies were selected. To eliminate the effect of MmP1 promoters or any potential genomic mutations, new plasmids containing *rpoD* mutants or wild-type *rpoD* driven by its native promoter were constructed and were transferred into *H. bluephagenesis* TD01Δenp. These *rpoD* mutants were analyzed through growth curve measurements. Among them, the best mutant (RpoD 3 M) contains three amino acid mutations: E35K, E181K, and M460I (Fig. 8d and Supplementary Fig. 27a). Further shake flask studies showed that the true cell mass (TCM) of RpoD 3 M mutant was about 84% higher than that of the wild-type RpoD strain (Fig. 8e and Supplementary Fig. 27b). Then, the domain prediction of RpoD revealed that it belongs to group 1 type of $\sigma$70 family, comprising four helical domains ($\sigma_{1.1}$, $\sigma_2$, $\sigma_3$, and $\sigma_4$). These three mutations are located in the $\sigma_{1.1}$, non-conserved region (NCR), and $\sigma_3$ domains of RpoD, respectively (Fig. 8f). The $\sigma_{1.1}$ domain functions as a gatekeeper, preventing $\sigma^{70}$ factor from binding non-specifically to promoter DNA when it is not bound to the core enzyme[51]. The NCR region is involved in core enzyme binding and promoter escape of $\sigma^{70}$ factor[52]. The $\sigma_3$ domain interacts with promoter DNA by specifically recognizing the extended −10 element[53]. Thus, RpoD 3 M may regulate the expression of multiple essential genes by modulating core enzyme and promoter binding, thereby conferring high tolerance to L-Arg.

## Evolution of *lysE* gene for higher L-Arg tolerance and efflux activity

LysE is an exporter in *Corynebacterium glutamicum* that transports both L-lysine (L-Lys) and L-Arg[54]. A recent study modified key residues of LysE by structural modeling and engineering, with A156V/L49T mutant demonstrating enhanced L-Arg extrusion[55]. Here, we accelerated mutation screening using orthogonal transcription mutators to obtain LysE mutants with enhanced L-Arg tolerance and efflux function. The codon-optimized *lysE* gene, driven by the constitutive P$_{porin42}$ promoter, was inserted into the endogenous plasmid as the target plasmid, with MmP1 promoter at both ends (Fig. 9a). The mutagenesis and screening process for LysE followed a similar procedure to that of RpoD. After two rounds of seeding to obtain LysE mutant libraries, these strains were transferred into 50MM medium with 5 g/L L-Arg for selection and spread onto 5 g/L L-Arg agar plates to screen for large clones. Subsequently, new plasmids without the MmP1 promoters were constructed to characterize the mutants. The best performer (LysE 6 M), with six amino acid mutations (Q31R, L38P, Y81R, T126A, L196P, and M235T), demonstrated a significantly higher OD$_{600}$ value and an approximately 48% improvement in TCM (Fig. 9b, c and Supplementary Fig. 28).

The three-dimensional structure of LysE indicates that these mutations are not located near the substrate binding sites, suggesting that they may indirectly affect extrusion activity via an allosteric mechanism (Fig. 9d). To further investigate the molecular mechanism underlying the enhanced activity of LysE 6 M, molecular docking was performed for both the mutant and wild-type LysE with L-Arg. From substrate and key residue interactions, the hydrogen bonding between LysE 6 M and L-Arg was enhanced. The interactions between D162 and L-Arg in LysE 6 M have one additional hydrogen bond compared to the wild type, potentially increasing binding stability (Fig. 9e–h). Subsequently, molecular dynamics simulations were conducted to further explore how LysE 6 M enhances extrusion activity. The results showed that the overall RMSF of LysE 6 M was significantly reduced, particularly in main loop region (T95-D137) and helical regions (L38-M94 and P157-L234) (Fig. 9i). The increased rigidity enhances the structural stability of the mutant. Additionally, LysE 6 M exhibited a more stable RMSD curve, reaching stabilization earlier at 16 ns (Fig. 9j). Moreover, LysE 6 M displayed a more stable radius of gyration (Rg) and lower overall potential energy, both of which are advantageous for binding and extrusion activity (Supplementary Fig. 29). These results demonstrated that the orthogonal transcription mutators enable efficient mutagenesis and evolution of the targeted protein in vivo.

## Discussion

In this study, we constructed orthogonal transcription mutators by fusing various cytosine and adenine deaminases with three phage RNAPs (MmP1, K1F, and VP4), generating both C:G to T:A and A:T to G:C mutations (Figs. 5, 6). These mutators were expressed on plasmids with broad-host inducible promoter P$_{Tac}$, enabling precise regulation of mutation frequencies via IPTG concentration. For instance, IPTG adjustments led to an over 20,000-fold increase in on-target mutation frequencies for pMT2-MmP1 and 3,300-fold for pMT3-MmP1 while

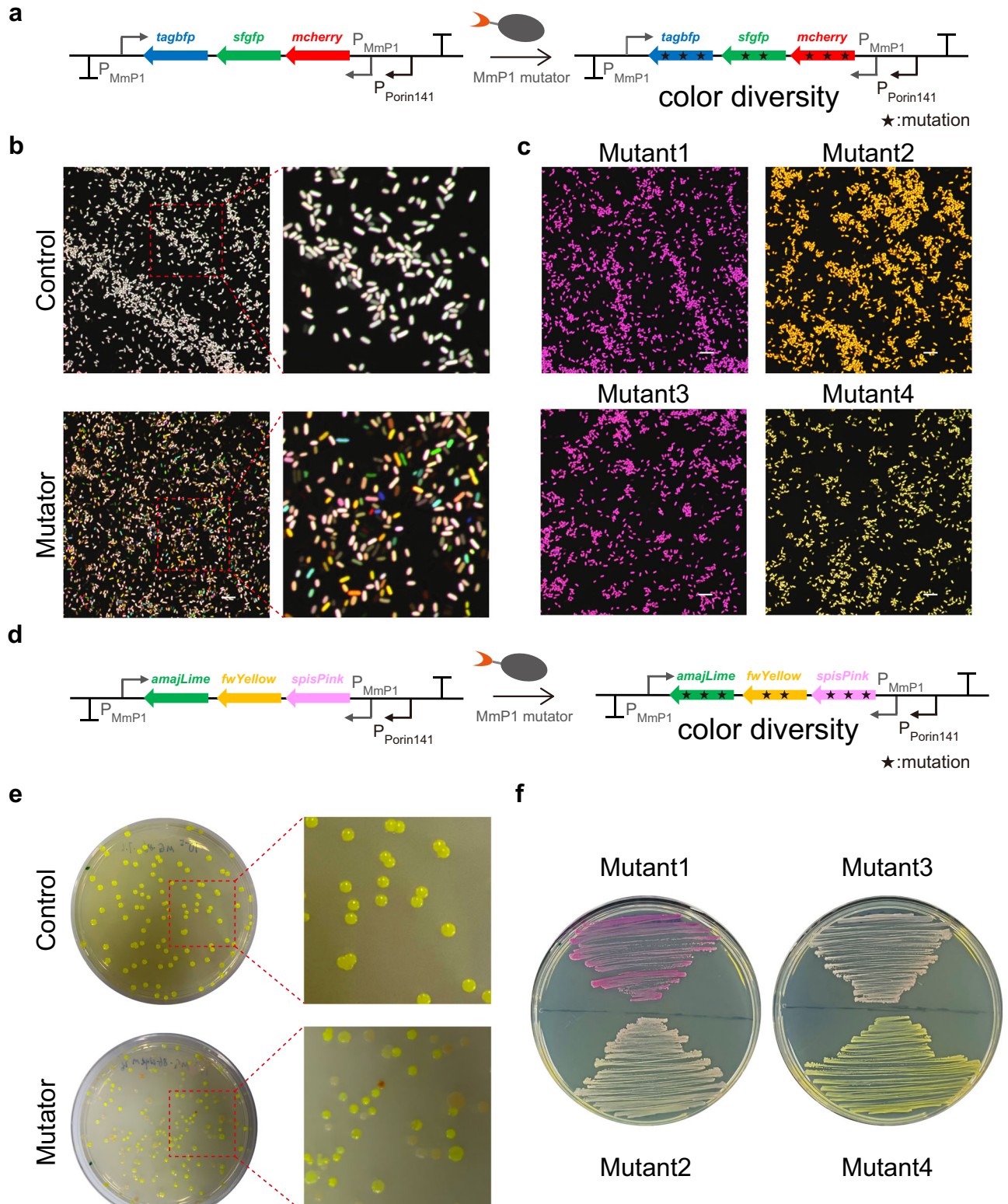

**Fig. 7 | Mutagenesis of fluorescent proteins and chromoproteins to generate diverse cellular colors in *E. coli*. a** Scheme of the mutagenesis process for the fluorescent proteins mCheery, sfGFP, and TagBFP, controlled by a strong constitutive promoter P$_{Porin141}$, involved the placement of two P$_{MmP1}$ promoters upstream and downstream of the gene cluster to recruit the MmP1-based dual type mutator. **b** Confocal microscopy analysis of the control and mutator groups in *E. coli*. The control group exhibited a white color, whereas the mutator group displayed a variety of colors like red, green, blue, and orange. Scale bar = 10 μm. The experiment was repeated three times with similar results. **c** Confocal microscopy analysis of fluorescent protein

mutants. mutants 1 and 3 exhibited purple, while mutants 2 and 4 displayed yellow. Scale bar = 10 μm. **d** Scheme of the mutagenesis process for chromoproteins amajLime, fwYellow, and spisPink, under P$_{Porin141}$ promoter, with two P$_{MmP1}$ promoters placed upstream and downstream of the gene cluster. **e** Observation of color variations in the mutator group in *E. coli*. The control group exhibited a green color, while the mutator group showed multiple colors, such as red and white. The experiment was repeated three times with similar results. **f** Chromoprotein mutant color profiles revealed that mutant 1 exhibited a purple color, mutants 2 and 3 were pink, while mutant 4 displayed a green color. Source data are provided as a Source Data file.

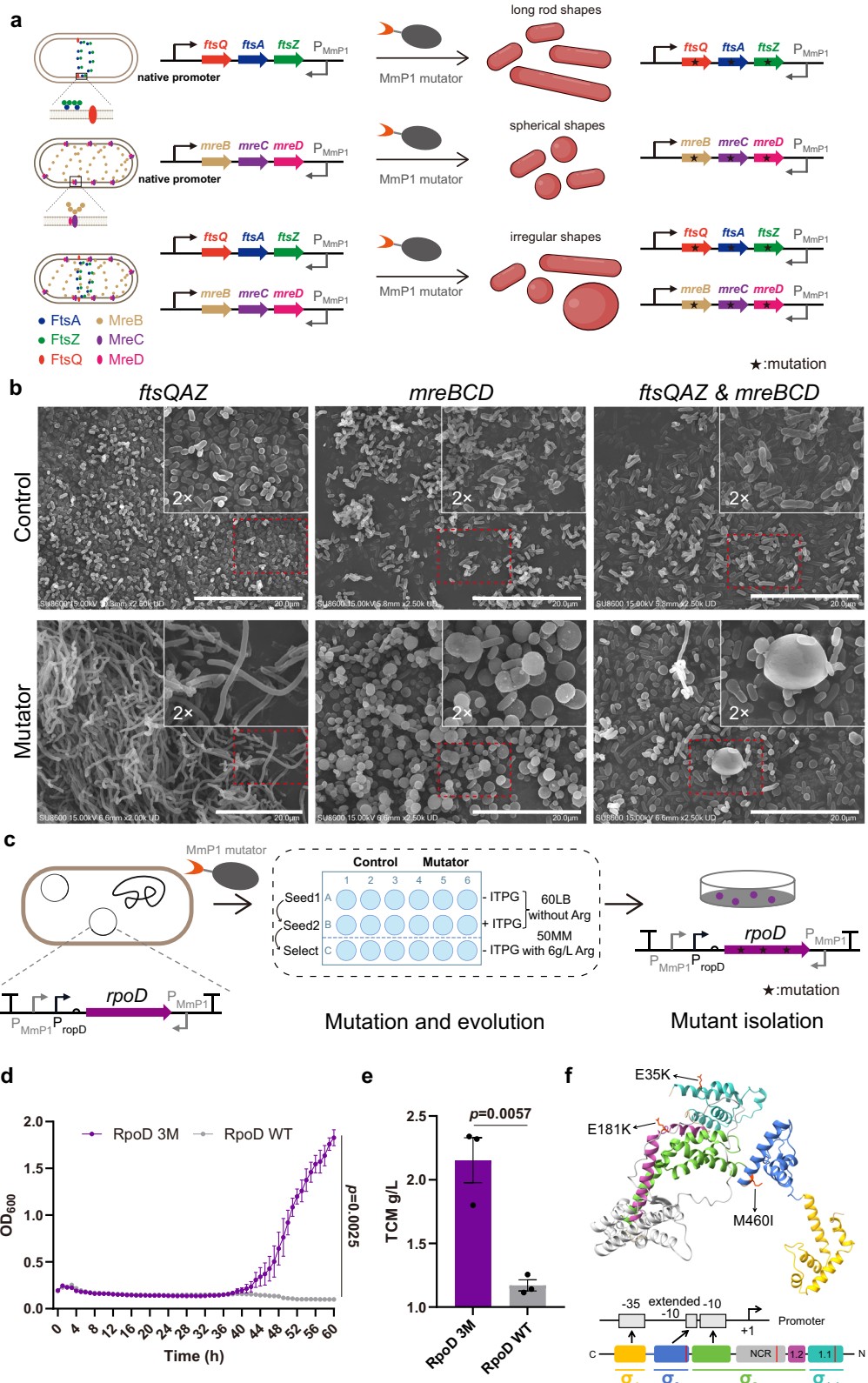

**Fig. 8 | Mutagenesis of cytoskeleton and cell division-related proteins for shape diversity, and sigma 70 factor RpoD for higher L-Arg tolerance in *H. bluephagenesis*. a** Scheme of the mutagenesis process for cytoskeleton and cell division-related proteins. The $P_{MmP1}$ promoters were placed downstream of *mreBCD* and *ftsQAZ* gene clusters in the reverse orientation. Arg represents L-arginine. **b** Scanning electron microscopy analysis of the shape changes in the control and mutator groups in *H. bluephagenesis*. The control group showed short rod shapes, while the mutator group revealed diverse shapes, including long rods and spheres. Scale bar = 20 µm. The experiment was repeated three times with

similar results. **c** Scheme of the evolution and selection procedure for σ^70 factor RpoD. RpoD was controlled by its native promoter, with MmP1 promoter at both ends. **d** Growth curve analysis of RpoD 3 M mutant and wild-type RpoD in 6 g/L L-Arg medium. **e** Shake flask results of RpoD 3 M mutant and wild-type RpoD in 6 g/L L-Arg medium. **f** Structure and domain analysis of RpoD, highlighting mutations in red. Data are presented as mean values and standard errors (error bars). *n* = 3, which represents three independent replicates of the experiment. Statistical analyses were performed using two-tailed Student's t-tests. A *p*-value < 0.05 was considered significant. Source data are provided as a Source Data file.

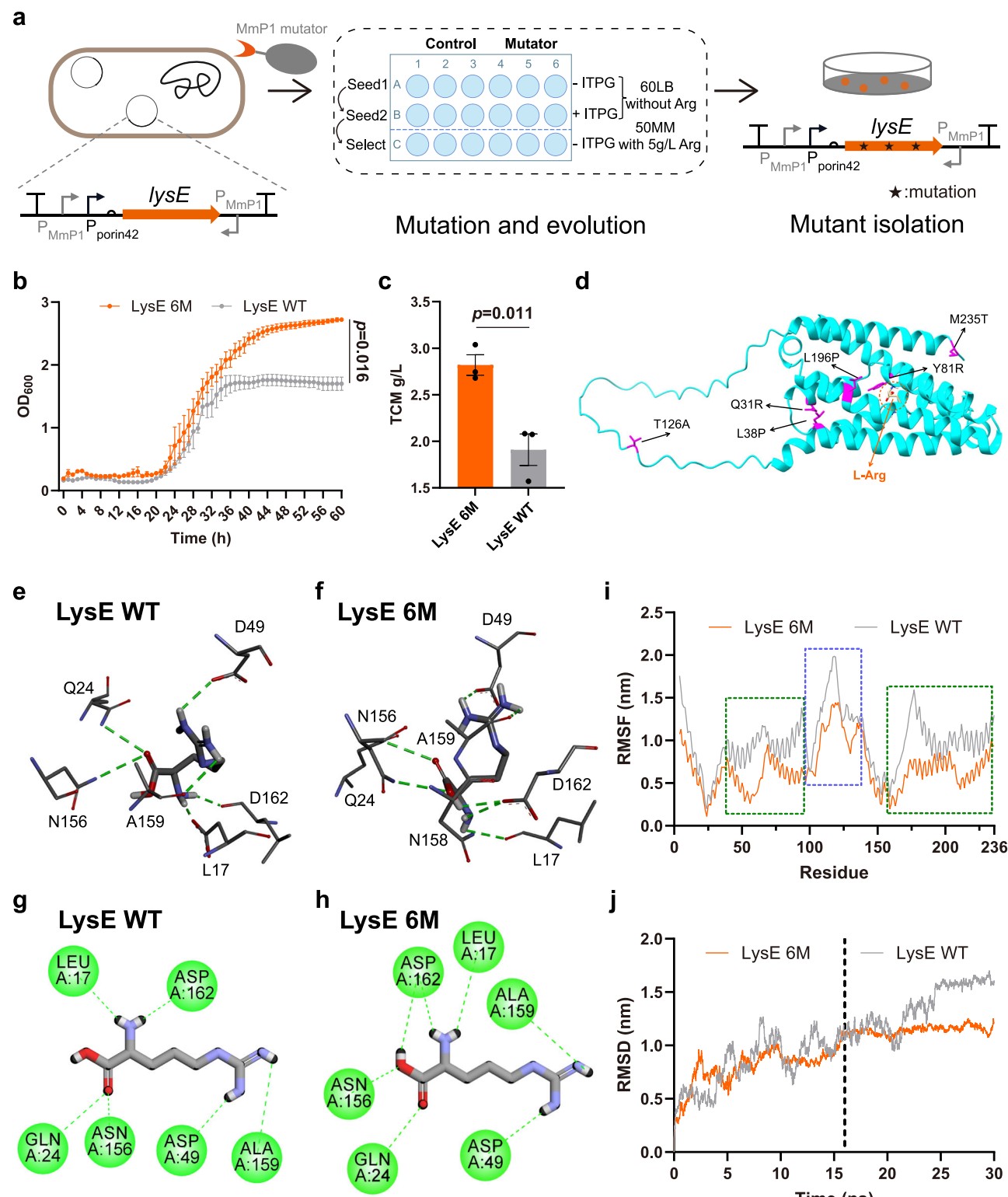

**Fig. 9 | Mutagenesis and evolution of LysE exporter for higher L-Arg tolerance.**
**a** Scheme of the evolution and selection procedure for LysE exporter. *lysE* gene is controlled by the constitutive $P_{porin42}$ promoter, with MmP1 promoter at both ends. Arg represents L-arginine. **b** Growth curve analysis of the LysE 6 M mutant and wild-type LysE in 5 g/L L-Arg 50MM medium. **c** Shake flask results of the LysE 6 M mutant and wild-type LysE in 5 g/L L-Arg 50MM medium. **d** Structure analysis of LysE, highlighting mutations in magenta. **e**, **f** Three-dimensional interaction of the LysE 6 M mutant and wild-type LysE with L-Arg. The green dotted lines represent hydrogen bonds. **g**, **h** Two-dimensional interaction of the LysE 6 M mutant and wild-type LysE with L-Arg. The green dotted lines represent hydrogen bonds. **i** The RMSF results of the LysE 6 M mutant and wild-type LysE with L-Arg. The loop region and the two helical regions are highlighted by blue and green boxes, respectively. **j** The RMSD results of the LysE 6 M mutant and wild-type LysE with L-Arg. Data are presented as mean values and standard errors (error bars). $n = 3$, which represents three independent replicates of the experiment. Statistical analyses were performed using two-tailed Student's t-tests. A $p$-value $< 0.05$ was considered significant. Source data are provided as a Source Data file.

minimizing cytotoxicity and off-target effects (Figs. 2, 4). In the future, appropriate inducer concentration can be selected based on the characteristics of the target proteins. The mutation rate of dual type mutator pMT23opt-MmP1 based on high-throughput sequencing results was 2.64 transition mutations/day/kb, higher than that of the previously reported MutaT7[transition][15] (1.58 transition mutations/day/kb), dual8 mutator[16] (0.46 transition mutations/day/kb), and MutaT7[GDE][16] (1.28 transition mutations/day/kb). In contrast to error-prone *PCR* and other in vitro methods[3], our in vivo platform avoids the time-intensive processes of library assembly and transformation, effectively overcoming the critical limitation of low transformation efficiency that restricts library diversity in traditional methods.

Phage RNAP fused with different deaminases exhibited varying mutagenesis activities, enhancing the flexibility and expandability of orthogonal transcription mutators. For instance, enhanced mutation activity was observed with deaminases like PmCDA1-UGI and TadA8e, outperforming counterparts such as evoPmCDA1-UGI and TadA7.10 (Fig. 2c, 4c). Additionally, the expression of UGI eliminates the need for *ung* gene knockout in the host genome, thereby improving the portability of orthogonal transcription mutators across different strains. Strategically combining cytosine and adenine deaminases enabled both C:G to T:A and A:T to G:C mutations. Recent studies reported modified TadA variants such as CABE T3.1, CABE T3.155, and TadDE, which show both adenine and cytosine mutation activity[56,57]. However, these TadA variant-MmP1 mutators produced only A:T to G:C mutations in *H. bluephagenesis* with no C:G to T:A mutation detected (Supplementary Fig. 30). This phenomenon may be due to differences in host cells (mammalian cells versus bacteria) and fusion strategies (Cas9 protein versus MmP1 RNAP).

We evaluated the off-target effects of orthogonal transcription mutators by quantifying mutations in the *rpoB* gene. The C:G to T:A type mutators exhibited low off-target rates (3 to 12-fold increase) (Figs. 2d, 3e), while the A:T to G:C type mutators displayed relatively higher off-target rates (122 to 253-fold increase) (Fig. 4d, k). For dual type mutators, pMT32d-MmP1 and pMT23opt-MmP1 demonstrated lower off-target rates, with 11- and 27-fold increases, respectively (Supplementary Fig. 8a). The elevated off-target rates of the A:T to G:C type mutators may be attributed to the high mutagenesis activity of TadA8e. In practical applications, adjusting inducer concentrations reduced off-target effects without compromising on-target rates. The off-target frequency of pMT23opt-MmP1 measured by the rifampicin assay was $2.6 \times 10^{-7}$, which is similar to that of MutaT7[transition], dual8, and MutaT7[GDE]. Therefore, mutators with low off-target effects, such as pMT32d-MmP1 and pMT23opt-MmP1, are preferable for mutagenesis and evolutionary applications.

DNA sequencing revealed that dual type mutators generated only base substitutions without insertions or deletions. In contrast, the control group exhibited many IS30 family transposon insertion mutations. Deactivating these transposons significantly reduced spontaneous mutation frequencies (Supplementary Fig. 9), indicating that intracellular transposons can cause genomic instability. Mutation hotspots were observed near phage promoters, likely due to RNAP detachment during transcription. However, positioning promoters upstream and downstream of the target gene resulted in an even distribution of mutations, a pattern also observed in MutaT7[14].

The orthogonal transcription mutators were developed using MmP1, K1F, and VP4, highlighting the versatility of this strategy across different phage RNAPs. Moreover, these mutators exhibited high mutation activity in the non-model *H. bluephagenesis* and the model organism *E. coli*, demonstrating broad-host capability, whereas the MutaT7 system is incompatible with *Halomonas* (Supplementary Figs. 11–15). Additionally, previous studies revealed that the MmP1, K1F, and VP4 RNAP expression systems perform well in other non-model organisms, including *P. entomophila*[21], *P. putida*[24], and *C. testosteroni*[25], suggesting potential for broader applications

with appropriate modifications. Furthermore, the mutators exhibited high specificity and orthogonality due to the unique phage RNAP-promoter pairs, enabling modular design for diverse mutagenesis applications.

Practical applications include rapid mutagenesis of fluorescent and chromoproteins for color variation, cytoskeleton and cell division-related proteins for morphology engineering, the sigma 70 factor RpoD and LysE exporter for higher arginine tolerance within a 24-hour mutagenesis period (Figs. 7–9). These evolutionary applications demonstrate that the orthogonal transcription mutation system can efficiently induce mutations in target proteins, generating mutants with desirable traits within one day of mutagenesis. Additionally, this system can target single genes, gene clusters, or multiple clusters simultaneously by rationally inserting phage promoters at locations, making it a versatile tool for rapidly evolving enzymes or enhancing metabolic pathways. While CRISPR-based approaches such as EvolvR[4] are generally limited to about 350 nt editing windows, our system enables efficient mutagenesis across substantially longer sequences (1.4 kb for *sacB* gene and 2.2 kb for fluorescent protein genes). In conclusion, the orthogonal transcription mutation system is an efficient, broad-host platform for 24 h mutagenesis and evolution, with great potential in laboratory evolution, enzyme modification, and metabolic engineering.

## Methods
### Strains, media, and cultivation conditions
The *Halomonas sp.* and *E. coli* strains used in this research are listed in Supplementary Table 1. *Halomonas bluephagenesis* TD01 was isolated from Aydingol Lake in China. *E. coli* MG1655 was utilized as the platform strain to characterize the orthogonal transcription mutators. *E. coli* S17-1 was used for plasmid construction and as a plasmid donor for conjugation. *E. coli* strain S17-1 and MG1655 were cultured at 37°C and 200 rpm in 10LB medium (10 g/L tryptone, 5 g/L yeast extract, and 10 g/L NaCl). *H. bluephagenesis* TD01 and its derivative strains were cultured at 37 °C and 200 rpm in 60LB medium (10 g/L tryptone, 5 g/L yeast extract, and 60 g/L NaCl). Solid media were made by adding 1.5–2% (w/v) agar powder to the corresponding liquid medium. The 20LB agar plate was used for conjugation (10 g/L tryptone, 5 g/L yeast extract, 20 g/L NaCl, and 1.5–2% w/v agar powder). The mineral salt medium (MM medium) was also used for *H. bluephagenesis* and its derivative strains and was adjusted to pH 8.0−9.0. For plasmid maintenance and selection, antibiotics were added to the medium at the following concentrations: 25 mg/L chloramphenicol (Cm), 50 mg/L kanamycin (Kan), 100 mg/L spectinomycin (Spe), 100 mg/L rifampicin (Rif), and 200 mg/L erythromycin (Em) in bacteria.

### Plasmid construction
The plasmids used in this study are listed in Supplementary Table 2. Relevant sequences can be found in Supplementary Table 3. DNA fragments were amplified by PCR using *high-fidelity Prime-Star DNA polymerase* (Takara Bio, Inc., Japan). Template plasmids were removed using the *DpnI* enzyme (New England Biolabs, Inc., USA). Then, DNA products were purified using the DNA Isolation Kit (Omega Bio-tek, Inc., USA). The plasmids were constructed using *Gibson* or Golden Gate assembly methods (New England Biolabs, Inc., USA) and transferred into *E. coli* S17-1 for conjugation or *E. coli* MG1655 for further characterization. The high-copy number plasmid pSEVA241 was used as the backbone for mutator plasmids, and the low-copy number plasmid pSEVA321 was used as the backbone for target plasmids[58].

### Conjugation and genome editing
Plasmids in *E. coli* S17-1 were transferred into *H. bluephagenesis* strains using the conjugation method. *E. coli* S17-1 carrying plasmid of interest, and *H. bluephagenesis* strains were cultivated in appropriate media with relevant antibiotics to OD600 of 0.6–0.8. Then, 500 μL cells from

each culture were mixed and harvested via centrifugation ($3000 \times g$ for 2 min). Cell precipitation was washed with 1 mL of 10LB medium and centrifuged again, and the supernatant was discarded. Next, the precipitation was resuspended in 35 µL of 10LB medium plated on the 20LB plates without antibiotics and incubated at 37 °C for 8–10 h. After mixing, the cells were placed on 60LB plates with appropriate antibiotics and kept at 37 °C for 2 days.

For genome editing, the CRISPR/Cas9 method was employed to insert genes of interest into the genome of *H. bluephagenesis* TD01 and *E. coli* MG1655 strains[59,60]. Briefly, *H. bluephagenesis* TD01 and *E. coli* MG1655 strains were transferred with a help plasmid expressing Cas9 and then transferred with a donor plasmid. Successful gene editing was validated by colony PCR and DNA sequencing.

## Mutation induction assay

For the erythromycin resistance gene recovery assay, *H. bluephagenesis* strains harboring a target plasmid expressing an inactivated ErmC mutant and different mutator plasmids were cultured in 60LB medium with Cm and Spe at 37 °C for 10 h. Next, the seed cultures were 100-fold diluted in 60LB medium containing 200 mg/L IPTG inducer (6.7 generations). After 20 h, the culture solution was diluted up to $10^7$-fold in 10-fold serial dilutions, spread on 60LB solid plates with or without 200 mg/L erythromycin, and incubated for 48 h. To calculate the off-target mutation rate, the culture solution was spread on 60LB solid plates containing 100 mg/L rifampicin.

For the *sacB* gene mutation assay, *H. bluephagenesis* strains with the *sacB* gene within the genome and *E. coli* MG1655 strains harboring a target plasmid with the *sacB* gene were incubated in 60LB medium with Spe and 10LB containing Cm and Spe, respectively, at 37 °C for 10 h. Then, the seed solution was diluted 100-fold in the culture medium containing 200 mg/L IPTG inducer and cultured at 37 °C for 20 h. Next, the solution from the culture was serially diluted in 10-fold steps and spread on 60LB or 10LB solid plates with or without 100 g/L sucrose, respectively. To assess the off-target rate, the culture solution was spread on solid plates with 100 mg/L rifampicin.

The on-target mutation frequency was calculated as the number of mutants grown on erythromycin or sucrose plates (CFU/mL)/number of total cells (CFU/mL). The off-target mutation frequency was calculated as the number of mutants grown on rifampicin plates/ number of total cells. The cell viability was assessed by the number of total cells. The Ma-Sandri-Sarkar (MSS) maximum likelihood method was utilized to calculate the mutation rate per base per generation (s.p.b.)[61]. Briefly, the phenotypic mutation rates were computed by dividing m values by the averaged cell numbers using the FluCalc website[62] (https://flucalc.ase.tufts.edu/). Then, the mutation rates per base per generation were calculated by dividing the phenotypic mutation rates by the copy number of target plasmid pSEVA321 (24, based on previous results[59]) and the number of ways ErmC mutants can reactivate (1 for the ErmC Y104S and Q10* mutants).

## Flow cytometry analysis

To verify whether the T7-like RNAP mutators can transcribe the target gene normally, the expression level of sfGFP in *H. bluephagenesis* strains was analyzed by flow cytometry. *H. bluephagenesis* strains harboring a target plasmid expressing sfGFP and a mutator plasmid were cultured in the 60LB medium containing Cm and Spe at 37 °C for 10 h. Then, the seed cultures were diluted 100-fold in the 50MM medium containing 200 mg/L IPTG inducer and incubated for 12 h. Subsequently, 2 µL of culture solution was diluted 100-fold in 200 µL of PBS for flow cytometry analysis (LSRFortessa4, BD Bioscience, USA). FITC (488 nm excitation light), FSC (forward scatter), and SSC (side scatter) channels were recorded. At least 10,000 cells were captured for each sample for subsequent analysis. Flow cytometry data were analyzed using FlowJo (v7.6) software.

## High-throughput sequencing and data analysis

To analyze the mutagenesis characteristics of the orthogonal transcription mutators, three dual-type mutators (pMT23d-MmP1, pMT32d-MmP1, and pMT23opt-MmP1), as well as the control pMT0-MmP1, were transferred into *H. bluephagenesis* strains with the *sacB* gene within the genome. Next, these strains were incubated in 60LB medium with 200 mg/L IPTG at 37 °C for 20 h for the mutation induction assay, as described above. Subsequently, to obtain DNA fragments containing the *sacB* gene, the culture solution was subjected to *PCR* using the primers 'CTGTTTGATGGTGGTTAACGGC' and 'CAGCGCCAGTAGTACTCCTATCA'. A total of a 3585 bp DNA fragment was amplified, containing a target region of 2024 bp, an upstream region of 961 bp, and a downstream region of 600 bp. The amplicon library construction, Illumina paired-end sequencing (PE150), and data preprocessing were performed by GENEWIZ, Inc. (Suzhou, China). The mutation rate was calculated as mutations per day per kb. Briefly, the mutation rate was calculated as the transition substitution frequency of the target region divided by the incubation time (days).

## Orthogonality studies

The orthogonality of three phage RNAP mutators was tested as follows: three *H. bluephagenesis* strains with the insertion of the *sacB* gene and a phage RNAP promoter in the genome, or three *E. coli* MG1655 strains containing a target plasmid expressing the *sacB* gene and phage RNAP promoters were used for study. Each strain was transformed with three RNAP dual type mutator plasmids and a control plasmid (expressing only phage RNAP), respectively. The subsequent mutation induction assay was the same as the steps described above. In this study, the mutation rate fold change was calculated as the mutation rate of the RNAP dual type mutator divided by the background mutation rate observed in the control group.

## Evolution and characterization of fluorescent protein and chromoproteins

For fluorescent protein and chromoprotein genes evolution in *E. coli* MG1655, the steps for evolution and characterization were as follows: the target plasmid containing *mcherry*, *sfgfp*, and *tagbfp* genes, and the target plasmid with *amajLime*, *fwYellow*, and *spisPink* genes, were each transformed into *E. coli* MG1655. The dual type mutator plasmid expressing MT23opt-MmP1 and a control plasmid expressing MmP1 RNAP were introduced into these two strains. These strains were cultured in the 10LB medium containing Cm and Kan at 37 °C for 10 h. The seed cultures were then diluted 100-fold in the 10LB medium supplemented with 200 mg/L IPTG inducer and incubated at 37 °C for 24 h. The culture solution was diluted $10^5$-fold, spread on 10LB solid plates with Cm, and incubated at 37 °C for 36 h. Colonies exhibiting color diversity were selected for colony PCR and DNA sequencing. The fluorescent diversity was further verified using a Multi-SIM AXR confocal super-resolution microscope (Nikon Instruments, Inc., Japan). 2 µL of culture was added to glass slides for observation under the confocal super-resolution microscope using a 60×oil immersion objective. The excitation wavelengths used were 405, 488, and 561 nm for TagBFP, sfGFP, and mCherry fluorescent proteins. Subsequently, the data were analyzed using NIS-Elements Viewer 5.21 software. The mutation and characterization steps of fluorescent protein genes in *H. bluephagenesis* strains were similar to those of *E. coli*, except that the 10LB medium was replaced with the 60LB medium. Kan was substituted with Spe to maintain the plasmids in *H. bluephagenesis* strains.

## Evolution and characterization of cytoskeleton and cell division-related proteins

For the evolution of cytoskeleton- and cell division-related genes in *H. bluephagenesis*, the procedure was as follows: The reverse MmP1

promoter was inserted downstream of the *mreBCD* cytoskeleton-related gene cluster and the *ftsQAZ* cell division-related gene cluster in the genome of *H. bluephagenesis* TD01, either separately (named TD01-*mreBCD*-P$_{MmP1}$, TD01-*ftsQAZ*-P$_{MmP1}$) or together (TD01-*mreBCD*&*ftsQAZ*-P$_{MmP1}$). The dual type mutator plasmid expressing MT23opt-MmP1 and a control plasmid expressing MmP1 RNAP were introduced into these three strains. These strains were cultured in the 60LB medium containing Spe at 37 °C for 10 h. The bacterial cultures were then diluted 100-fold in 60LB medium supplemented with 200 mg/L IPTG inducer and incubated at 37 °C for 24 h. To avoid the impact of the dual type mutator MT23opt-MmP1 or MmP1 RNAP on native gene expression, the culture was diluted 1:100 into 60LB medium without IPTG inducer and incubated at 37 °C for 24 h. The resulting samples were harvested via centrifugation (3000×g for 2 min) for further observation using the Olympus IX83 inverted microscope and ultrahigh-resolution scanning electron microscope SU8600 (Hitachi Ltd, Japan). The mutation and characterization protocol of cytoskeleton- and cell division-related genes in *E. coli* MG1655 strains was similar to that of *H. bluephagenesis*, except that 60LB medium was replaced with 10LB medium, and Spe was substituted with Kan to maintain the plasmids in *E. coli* MG1655 strains.

### Evolution and characterization of sigma 70 factor RpoD and exporter LysE

The target plasmids (pMT91 or pMT94), the dual mutator plasmid expressing pMT23opt-MmP1, and the control plasmid pMT0-MmP1 were introduced into *H. bluephagenesis* TD01Δ*enp* via conjugation. To construct the target plasmid pMT91 containing codon-optimized *lysE* gene from *Corynebacterium glutamicum* was driven by Pporin42 promoter with MmP1 promoter at both ends. For the target plasmid pMT94 containing sigma factor RpoD, the *rpoD* gene from *H. bluephagenesis* was driven by its native upstream promoter with MmP1 promoter at both ends. Three colonies were inoculated in 60LB medium containing Cm and Spe at 37 °C for 10 h as the first seed culture. Subsequently, the bacterial culture was diluted 100-fold in 60LB medium with 200 mg/L IPTG inducer and incubated at 37 °C for 24 h as the second seed culture to facilitate mutation accumulation. Next, the culture was diluted 100-fold in 50MM medium containing 5 g/L or 6 g/L L-arginine (L-Arg) for 24 h to select the evolved mutants. The entire culture process was conducted in a 96-deep well plate. At the end of the evolution process, the cell culture was spread on 50MM solid plates containing 5 g/L or 6 g/L L-arginine and Cm. After cultivation at 37 °C, large colonies were selected from the plates for colony PCR and sequencing. New plasmids containing *lysE* or *rpoD* mutant genes, driven by P$_{porin42}$ promoter or native *rpoD* promoter without MmP1 promoters, were then constructed and transferred into *H. bluephagenesis* TD01Δ*enp* to eliminate the effects of MmP1 promoter or genomic mutations. The wild-type *lysE* and *rpoD* genes were used as control group. The growth conditions of the mutants were assessed through growth curve measurements and shake flask studies under 5 g/L or 6 g/L L-Arg 50MM medium.

### Cell growth characterization

The cell growth characterization of LysE or RpoD mutants was determined using cell growth curve measurements and shake flask studies. For shake flask experiments, single colonies from the 60LB plates were transferred into 20 mL of 60LB medium and incubated at 37 °C, 200 rpm for 12 h. The seed culture was then diluted 100-fold in 60LB medium and cultured at 37 °C, 200 rpm for 10 h. Finally, 1 mL of the culture solution was inoculated into 20 mL of 50MM medium with 5 g/L or 6 g/L L-arginine in a 100 mL conical flask and incubated at 37 °C, 200 rpm for 48 h. Each group had three replicates. After the shake flask experiment, 20 mL of the culture solution was centrifuged at 8000 × g for 5 min, then resuspended with 20 mL of distilled water

and centrifuged again to remove the salt. The cell dry weight (CDW) of each sample was measured after 24 h of lyophilization. Then, PHA content was determined by gas chromatography (GC-2014, SHIMADZU, Japan), and true cell mass (TCM) was calculated as CDW minus PHA content, representing the cell growth condition. The seed solution preparation process of cell growth curve experiment was similar to the shake flask experiment. After seed solution preparation, seed cultures were diluted 100-fold in 50MM medium with 5 g/L or 6 g/L L-arginine in a 96-well plate and incubated at 37 °C, 200 rpm for 60 h. OD$_{600}$ values were measured every hour using a BioTek SYNERGY H1 microplate reader (BioTek Instruments, Inc., USA).

### Structure analysis, molecular docking, and MD simulations

The three-dimensional structures of LysE, RpoD, and their mutants were predicted using AlphaFold2.0[63]. Molecular docking of small molecule ligand (L-Arg) and the receptor protein LysE was performed using AutoDock Vina 1.2.2[64], and the docking results were visualized using ChimeraX 1.8 software[65]. Molecular dynamics simulations were carried out using GROMACS 2024.1 software for 30,000 ps (30 ns) at 300 K and 1 atm, employing the CHARMM36 force field[66,67]. Domain prediction of RpoD from *H. bluephagenesis* was performed using InterPro[68] (https://www.ebi.ac.uk/interpro/).

### Statistics

The experimental data were presented as means and standard errors, with at least three independent replicates, and analyzed using Prism v8 software (GraphPad Software, Inc., USA). Statistical significance was defined as $P < 0.05$.

### Reporting summary

Further information on research design is available in the Nature Portfolio Reporting Summary linked to this article.

## Data availability

There is no restriction on the data associated with this study. DNA sequencing data generated in this study have been deposited in the NCBI Sequence Read Archive database under accession code PRJNA1272322. All data related to this study can be found in the main manuscript and the supplementary materials. Source data are provided with this paper.

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

## Acknowledgements

Funding: This work was financially supported by grants from the National Natural Science Foundation of China (Grant No. 32130001 to G.Q.C.) and Tsinghua-Peking University Center of Life Sciences. We are very grateful to Prof. Victor de Lorenzo of CSIC/Spain for kindly donating the pSEVA series plasmids.

## Author contributions

Conceptualization: G.Q.C., M.W.S. Methodology: M.W.S., Z.N.Z., X.F.J., J.D. Investigation: M.W.S. Formal analysis: M.W.S. Validation: M.W.S. Visualization: M.W.S. Funding acquisition: G.Q.C. Supervision: G.Q.C. Writing—original draft: M.W.S. Writing—review & editing: G.Q.C., M.W.S., Z.N.Z., X.F.J., J.D.

## Competing interests

The authors declare no competing interests.
