## [Peer Review File · Nature Communications]

An Orthogonal Transcription Mutation System Generating All Transition Mutations for Accelerated Protein Evolution in vivo

Corresponding Author: Professor Guo-Qiang Chen

Version 0:

Reviewer comments:

Reviewer #1

(Remarks to the Author)

The authors achieved highly efficient mutagenesis non-model organism *Halomonas bluephagenesis* and model organism *Escherichia coli* based on orthogonal transcription element MmP1, K1F, and VP4 RNA polymerases. First, the authors established a system capable of achieving C:G >T:A base editing by utilizing PmCDA1, UGI, and MmP1 RNA polymerase and optimized the expression level of UGI to enhance the mutation efficiency. Subsequently, the authors achieved A:T >G:C mutations by fusing various TadA mutants to the MmP1 RNA polymerase mutant. Next, the authors combined the two mutation systems, enabling the simultaneous introduction of both types of mutations and verifying their orthogonality. Finally, they demonstrated the practical utility of the tool through evolution experiments using different metabolic and morphology engineering.

Overall, this study established orthogonal transcription-based continuous evolution systems based on MmP1, K1F, and VP4 RNA polymerases. The work is well-organized and methodologically sound, offering practical insights for designing high-yield biocatalysis system. Compared with the MutaT7-based continuous evolution system based on commonly used T7 RNA polymerase, novel identified MmP1, K1F, and VP4 RNA polymerases have their unique advantages as key genetic component for driving continuous evolution, especially for smaller molecular weight for efficient assembly with base editor, which expands toolbox for continuous evolution.

The following minor concerns needs to be addressed:

1. The method used by the authors to measure mutation rates—employing a target plasmid expressing an inactivated ErmC—may be not accurate enough. By calculating the mutation rate as the ratio of erythromycin-resistant cells to the total number of cells within a given population, the authors overlook critical factors such as the types of restoration mutations, the number of cell generations, and plasmid copy number, all of which can influence the calculated mutation rate. To accurately determine the mutation rate, it is necessary to use a maximum likelihood function and normalize the results by accounting for the types of restoration mutations, the number of cell generations, and plasmid copy number. Using established tools for calculating mutation rates, such as FALCOR (Fluctuation Analysis Calculator with Optional Recombination) or FluCalc (<https://flucalc.ase.tufts.edu/>) to double check the mutation rate may be helpful.
2. Analyzing mutation distribution, calculate mutation rates, and characterize the mutation spectrum based on high-throughput sequencing would provide overview of the efficiency of the newly developed tool for continuous evolution.
3. Specifying the unique component or properties in the title may clearly convey key information for the readers, such as the mechanism of mutation (orthogonal transcription) and the functional outcomes of the mutations (C:G >T:A and A:T >G:C).
4. Statistical significance analysis is unnecessary for Figure 2. f–h and Figure 4. f–h.
5. In Figure 5, the panel labels are not arranged in the order of their first appearance (e.g., panel "k" appears earlier than it should). Line 245, "(Fig. 51)" seems to be a typo and should likely be corrected to "(Fig. 5l)."

Reviewer #2

(Remarks to the Author)

Polymerase-guided base editing enables in vivo mutagenesis and rapid protein engineering. Many in vivo polymerase-guided base editing tools relying on T7 RNA polymerase (such as MutaT7, eMutaT7) have been reported so far, because the transcription of T7 RNAP is orthogonal with host RNAP. This strategy has already been applied in *E. coli*, *Corynebacterium glutamicum*, mammals, yeast and plants. Besides, the in vivo mutagenesis tools not dependent on T7 RNAP have also been reported in *E. coli* (doi: 10.1093/nar/gkab1244).

Non-model organism *Halomonas bluephagenesis* TD01 was isolated by the authors, which has been industrially used in PHA production. However, the commonly used T7 system failed to adapt to *Halomonas* sp. TD01 for several unknown reasons. The authors characterized three T7-like systems (Mmp1, VP4 and K1F) for *Halomonas* sp. TD01 in a previous study (<https://doi.org/10.1016/j.ymben.2016.11.007>). The T7-like systems were mined from the phage genome database, so they were orthogonal with bacterial endogenous RNAP, which functioned like T7 RNAP.

The authors fused Mmp1, K1F, and VP4 T7-like RNAP with deaminases, and then carried out condition optimizations to obtain mutators for *Halomonas* sp. TD01. Finally, the mutations of chromoproteins were demonstrated in this work. It can be seen that the fusion of T7-like RNA polymerase and deaminase to construct the targeted mutagenesis system for *Halomonas* sp. TD01 is exactly the same as the design idea of MutaT7, so the novelty of this manuscript is somewhat weak. So the reviewer thinks this work is insufficient to meet the high standard of Nature Communications, and journals specific to nucleic acid may be more appropriate.

Major Comments

1. The authors should compare the system developed in this work with other reported targeted mutagenesis systems, including mutation rates, off-target rates, applicable scene and so on.
2. The authors employed this system to mutate chromoproteins. However, *Halomonas* sp. TD01 has been used as a representative industrial strain, it is suggested that the authors should employ this system to one or even more important industrial enzymes or chemical production to demonstrate the importance of developing this system for *Halomonas* sp. TD01.

Minor Comments

The text or letters in some figures is too small, such as Fig.5c, Fig 5g, Fig 5k, Fig 4a.

Reviewer #3

(Remarks to the Author)

Reviewer comments on "An Ultrahigh Efficient Mutation System for in vivo Mutagenesis and Evolution":

- 1) How did the authors calculate the "1,500,000-fold increase in mutation rate"?
- 2) How did the authors monitor the mutation rate of "muta-T7"?
- 3) The authors calculated the "evolution of genes encoding fluorescent proteins", but does this method apply similarly to other genes?
- 4) What is the breakthrough of this study compared to previous research, particularly regarding "deaminase"?
- 5) Please discuss prior methods (e.g., CRISPR-based mutagenesis, error-prone PCR, etc.) in comparison to your results.
- 6) Finally, can this toolkit be applied to a eukaryotic system?

Version 1:

Reviewer comments:

Reviewer #1

(Remarks to the Author)

The reviewer's concerns have been properly addressed.

Reviewer #2

(Remarks to the Author)

The authors have addressed all previous comments, and the revised manuscript shows clear improvement in clarity and quality.

I find the revisions appropriate and consider the manuscript suitable for publication.

Reviewer #3

(Remarks to the Author)

Accepted after revision.

Response to Reviewer Comments

We would like to thank all of the reviewers for their feedback. We appreciate their insightful comments and suggestions, which helped us to strengthen the manuscript. We have carefully incorporated these into our revised manuscript and addressed each comment made by the reviewers in details (in blue font). All modifications or additional data are highlighted in red in the revised manuscript.

Reviewer #1 (Remarks to the Author):

The authors achieved highly efficient mutagenesis non-model organism *Halomonas bluephagenesis* and model organism *Escherichia coli* based on orthogonal transcription element MmP1, K1F, and VP4 RNA polymerases. First, the authors established a system capable of achieving C:G >T:A base editing by utilizing PmCDA1, UGI, and MmP1 RNA polymerase and optimized the expression level of UGI to enhance the mutation efficiency. Subsequently, the authors achieved A:T >G:C mutations by fusing various TadA mutants to the MmP1 RNA polymerase mutant. Next, the authors combined the two mutation systems, enabling the simultaneous introduction of both types of mutations and verifying their orthogonality. Finally, they demonstrated the practical utility of the tool through evolution experiments using different metabolic and morphology engineering.

Overall, this study established orthogonal transcription-based continuous evolution systems based on MmP1, K1F, and VP4 RNA polymerases. The work is well-organized and methodologically sound, offering practical insights for designing high-yield biocatalysis system. Compared with the MutaT7-based continuous evolution system based on commonly used T7 RNA polymerase, novel identified MmP1, K1F, and VP4 RNA polymerases have their unique advantages as key genetic component for driving continuous evolution, especially for smaller molecular weight for efficient assembly with base editor, which expands toolbox for continuous evolution.

We sincerely appreciate the reviewer's positive evaluation and thoughtful summary of our work. Thank you for your careful reading and constructive feedback. Below, we have addressed the reviewer's concerns in details:

The following minor concerns needs to be addressed:

1. The method used by the authors to measure mutation rates—employing a target plasmid expressing an inactivated ErmC—may be not accurate enough. By calculating the mutation rate as the ratio of erythromycin-resistant cells to the total number of cells within a given population, the authors overlook critical factors such as the types of restoration mutations, the number of cell generations, and plasmid copy number, all of which can influence the calculated mutation rate. To accurately determine the mutation rate, it is necessary to use a maximum likelihood function and normalize the results by accounting for the types of restoration mutations, the number of cell generations, and plasmid copy number. Using established tools for calculating

mutation rates, such as FALCOR (Fluctuation Analysis Calculator with Optional Recombination) or FluCalc (<https://flucalc.ase.tufts.edu/>) to double check the mutation rate may be helpful.

We appreciate your valuable suggestions. Based on your suggestion, we have recalculated the mutation rates per base per generation for the erythromycin recovery experiments by incorporating other key parameters^{1,2}. Additionally, as recommended, we reassessed the mutation rates using maximum likelihood estimation via FluCalc^{3,4}. These updates have been integrated into the Results section (Pages 7-10, Lines 124-126, 151-156, and 181-182). The detailed methodological descriptions provided in the Methods section (Page 29, Lines 561-567).

2. Analyzing mutation distribution, calculate mutation rates, and characterize the mutation spectrum based on high-throughput sequencing would provide overview of the efficiency of the newly developed tool for continuous evolution.

We sincerely appreciate your constructive suggestions. High-throughput sequencing can indeed better assess the mutation characteristics of orthogonal transcription mutators. Therefore, we conducted NGS sequencing for three MmP1-based orthogonal transcription mutators (pMT23d-MmP1, pMT32d-MmP1, and pMT23opt-MmP1) to reassess mutation distributions, mutation rates, and mutation types. High-throughput sequencing results are consistent with our earlier findings from mutation induction assays and Sanger sequencing (Figs. 5b, 5l). These results have been incorporated into in the Results section (Pages 13-14, Lines 255-276, new Figs. 6a-6f and new Supplementary Fig. 10).

3. Specifying the unique component or properties in the title may clearly convey key information for the readers, such as the mechanism of mutation (orthogonal transcription) and the functional outcomes of the mutations (C:G >T:A and A:T >G:C).

We gratefully acknowledge this insightful suggestion. We have revised the title to “An Ultrahigh Efficient Orthogonal Transcription Mutation System Generating All Transition Mutations for Specific Mutagenesis and Evolution *in vivo*” to better clarify the main points of our work.

4. Statistical significance analysis is unnecessary for Figure 2. f-h and Figure 4. f-h.

We thank you for the comment and have removed the significance analysis for these figures as suggested.

5. In Figure 5, the panel labels are not arranged in the order of their first appearance (e.g., panel "k" appears earlier than it should). Line 245, "(Fig. 5l)" seems to be a typo and should likely be corrected to "(Fig. 5l)."

We sincerely appreciate your careful reading on the details of the article and apologize for the oversight. We have now reorganized the panel labels in Figure 5 to match their order of appearance in the text and modify the corresponding panel labels in the Results section (Page 13, Lines 249-254). Additionally, the typo error has been corrected to “(Fig. 5I)”.

Reviewer #2 (Remarks to the Author):

Polymerase-guided base editing enables *in vivo* mutagenesis and rapid protein engineering. Many *in vivo* polymerase-guided base editing tools relying on T7 RNA polymerase (such as MutaT7, eMutaT7) have been reported so far, because the transcription of T7 RNAP is orthogonal with host RNAP. This strategy has already been applied in *E. coli*, *Corynebacterium glutamicum*, mammals, yeast and plants. Besides, the *in vivo* mutagenesis tools not dependent on T7 RNAP have also been reported in *E. coli* (doi: 10.1093/nar/gkab1244).

Non-model organism *Halomonas bluephagenesis* TD01 was isolated by the authors, which has been industrially used in PHA production. However, the commonly used T7 system failed to adapt to *Halomonas* sp. TD01 for several unknown reasons. The authors characterized three T7-like systems (MmP1, VP4 and K1F) for *Halomonas* sp. TD01 in a previous study (<https://doi.org/10.1016/j.ymben.2016.11.007>). The T7-like systems were mined from the phage genome database, so they were orthogonal with bacterial endogenous RNAP, which functioned like T7 RNAP.

The authors fused MmP1, K1F, and VP4 T7-like RNAP with deaminases, and then carried out condition optimizations to obtain mutators for *Halomonas* sp. TD01. Finally, the mutations of chromoproteins were demonstrated in this work. It can be seen that the fusion of T7-like RNA polymerase and deaminase to construct the targeted mutagenesis system for *Halomonas* sp. TD01 is exactly the same as the design idea of MutaT7, so the novelty of this manuscript is somewhat weak. So the reviewer thinks this work is insufficient to meet the high standard of Nature Communications, and journals specific to nucleic acid may be more appropriate.

We sincerely appreciate your thorough evaluation and valuable suggestions. We have carefully addressed each of the reviewer’s comments in details below. We hope our responses and revisions could meet the reviewer’s expectations and have significantly improved the manuscript.

Regarding the point that “It can be seen that the fusion of T7-like RNA polymerase and deaminase to construct the targeted mutagenesis system for *Halomonas* sp. TD01 is exactly the same as the design idea of MutaT7, so the novelty of this manuscript is somewhat weak”.

We appreciate your concerns and welcome this opportunity to clarify the novelty of our work. Our orthogonal transcription mutation system demonstrates three key advances over existing technologies:

(1) Higher mutagenesis efficiency

High-throughput sequencing revealed that the mutation rate of pMT23opt-MmP1, which can produce all transition mutations, was 2.64 transition mutations/day/kb. In comparison, the mutation rate of MutaT7^{transition5} was 1.58 mutations/day/kb, the dual⁸⁶ mutator was 0.46 mutations/day/kb, and MutaT7^{GDE6} was 1.28 mutations/day/kb, based on their high-throughput sequencing results. We acknowledge that enhanced mutagenesis efficiency enables faster sampling of the proteins' fitness landscape, which can facilitate the generation of diverse and functionally improved mutants in a short time, as we have shown in our applications.

(2) Broader applicability

Orthogonal transcription mutators could function effectively in both non-model organism *Halomonas bluephagenesis* and model organism *E. coli*, whereas the MutaT7 system is incompatible with *Halomonas*. To our knowledge, this is the first report of an efficient targeted evolution system developed for extremophiles. Additionally, orthogonal transcription mutators based on three phage RNA polymerases exhibited strong orthogonality. This property facilitates modular design and provides opportunities for diverse applications in future research. Therefore, we believe that our system complements and broadens the scope of the MutaT7 system, especially for non-model organisms with significant biotechnological potential.

(3) The desired mutants could be obtained within only one day of mutation evolution

Since orthogonal transcription mutators demonstrate high mutation efficiency, we believe that 24 hours of mutagenesis is sufficient to build rich and diverse targeted protein mutant libraries. Therefore, we did not perform continuous evolution of the targeted protein through successive passages, as commonly found in the MutaT7 system. Instead, we simply mutated the targeted protein for 24 hours, followed by desired mutant screening. As demonstrated by mutagenesis of fluorescent proteins, chromoproteins, cytoskeleton and cell division-related proteins, a 24-hour mutagenesis period was sufficient to induce diverse color and morphological variations. Additionally, the global transcription factor RpoD and the LysE exporter were subjected to 24-hour mutagenesis, followed by screening of RpoD and LysE mutants capable of tolerating high arginine concentrations. These successful applications confirm the efficiency of this system.

Major Comments

1. The authors should compare the system developed in this work with other reported targeted mutagenesis systems, including mutation rates, off-target rates, applicable scene and so on.

Thank you for your insightful suggestions. We compared the orthogonal transcription mutators with the reported MutaT7 systems and included them in the Discussion

section (Pages 23-26, Lines 440-447, 468-470, and 481-484), and would like to summarize the key points here:

(1) Our system has higher mutation efficiency. We recalculated the mutation rate of dual type mutator pMT23opt-MmP1 based on high-throughput sequencing results, which was 2.64 transition mutations/day/kb. The mutation rate is higher than that of the previously reported MutaT7^{transition}⁵ (1.58 mutations/day/kb), dual8⁶ mutator (0.46 mutations/day/kb), and MutaT7^{GDE}⁶ (1.28 mutations/day/kb).

(2) The off-target frequency is comparable to that of the previous MutaT7 system. The off-target frequency of pMT23opt-MmP1 measured by the rifampicin assay was 2.6×10^{-7} , which is similar to that of MutaT7^{transition}, dual8, and MutaT7^{GDE}.

(3) Our system has broader applicability. Orthogonal transcription mutators show high efficiency in both the non-model organism *H. bluephagenesis* and the model organism *Escherichia coli*, whereas the MutaT7 system is incompatible with *Halomonas*. Additionally, the strong orthogonality exhibited by orthogonal transcription mutators based on three phage RNA polymerases facilitates modular design for diverse applications in future research.

2. The authors employed this system to mutate chromoproteins. However, *Halomonas* sp. TD01 has been used as a representative industrial strain, it is suggested that the authors should employ this system to one or even more important industrial enzymes or chemical production to demonstrate the importance of developing this system for *Halomonas* sp. TD01.

We sincerely appreciate your constructive suggestions. We applied orthogonal transcription mutators to the sigma 70 factor RpoD and the LysE exporter to enhance arginine tolerance and efflux activity in *Halomonas bluephagenesis*.

(1) Global transcription machinery engineering (gTME) has been widely used to modify cell factories for enhanced tolerance or high-yield mutants^{7, 8}. Using orthogonal transcription mutators, we performed *in vivo* mutagenesis of RpoD and successfully identified arginine-tolerant RpoD mutants after just 24 hours of mutation, bypassing the requirement for successive passages. The best mutant (RpoD 3M) contains three amino acid mutations: E35K, E183K, and M460I, and it exhibits a higher OD₆₀₀ value and true cell mass (Pages 19-21, Lines 364-400, new Figs. 8 c-f).

(2) We employed orthogonal transcription mutators to evolve LysE (a key arginine exporter), and identified the LysE 6M mutant with improved arginine tolerance and efflux function within a 24-hour mutagenesis period. LysE 6M mutant contains 6 amino acids substitutions: Q31R, L38P, Y81R, T126A, L196P, and M235T. Notably, molecular docking and MD simulations revealed that these six mutant residues enhance the binding affinity for arginine not through direct interaction, but rather via indirect structural stabilization effects. Thus, we propose that orthogonal transcription mutators can effectively discover complex functional optimizations to identify functionally optimized mutants (Pages 21-22, Lines 401-431, new Fig. 9).

Therefore, these applications across multiple protein classes support the potential generality of our system for diverse targeted proteins for industrial purposes.

Minor Comments

The text or letters in some figures is too small, such as Fig.5c, Fig 5g, Fig 5k, Fig 4a.

We sincerely appreciate your careful examination of our figures. We apologize for the oversight regarding the small text size in these. Following the reviewer's suggestion, we have carefully revised these figures by enlarging text elements and labels to ensure optimal readability. The modified figures are included in the revised manuscript.

Reviewer #3 (Remarks to the Author):

Reviewer comments on “An Ultrahigh Efficient Mutation System for *in vivo* Mutagenesis and Evolution”:

1) How did the authors calculate the “1,500,000-fold increase in mutation rate”?

We sincerely appreciate your question regarding the calculation of the mutation rate increase. The 1,500,000-fold increase was determined by comparing the mutation rates per base per generation between our mutator pMT2-MmP1-W and the control. The Ma-Sandri-Sarkar (MSS) maximum likelihood method was utilized to calculate the mutation rate per base per generation³. The mutation rate of pMT2-MmP1-W is 3.9×10^{-4} substitutions per base (s.p.b.) while the control is 2.6×10^{-10} s.p.b. Therefore, the fold-increase was calculated as: $3.9 \times 10^{-4} / 2.6 \times 10^{-10} \approx 1,500,000$ -fold. The corresponding values were presented in the revised manuscript (Page 8, Lines 151-153). We appreciate the opportunity to clarify this important detail.

2) How did the authors monitor the mutation rate of “muta-T7”?

We thank you for your question regarding our mutation rate monitoring methodology. To comprehensively evaluate the mutation rate in our system, we employed two experimental approaches (Pages 28-29, Lines 543-557):

(1) Erythromycin resistance recovery assay: This method involved culturing strains containing the inactivated ErmC mutant, followed by IPTG induction and plating on both selective (with erythromycin) and non-selective media to accurately quantify functional revertants.

(2) *sacB* gene mutation assay: We cultured strains carrying the sucrose-sensitive *sacB* gene, induced mutations with IPTG, and then plated on sucrose-containing and control media to assess loss-of-function mutations or performed PCR for high-throughput sequencing.

3) The authors calculated the “evolution of genes encoding fluorescent proteins”, but does this method apply similarly to other genes?

We appreciate your insightful question regarding the general applicability of our mutagenesis method. We are truly grateful for your thoughtful inquiry, which allows us to better highlight the versatility of our system. Indeed, our orthogonal transcription mutator system has demonstrated remarkable efficiency in evolving not just fluorescent proteins, but also several other functionally diverse proteins:

- (1) Cytoskeletal and cell division-related proteins resulting in distinct morphological variants (Pages 18-19, Lines 335-363, new Figs. 8a, 8b);
- (2) The global transcription factor RpoD with enhanced arginine tolerance (Pages 19-21, Lines 364-400, new Figs. 8c-8f) (additional study);
- (3) The LysE exporter with improved efflux function (Pages 21-22, Lines 401-431, new Fig. 9) (additional study);

Notably, the high mutation efficiency of our system enables the generation of rich, diverse mutant libraries within just 24 hours of mutagenesis. The applications across these diverse proteins strongly suggest broad applicability to other gene products as well.

4) What is the breakthrough of this study compared to previous research, particularly regarding “deaminase”?

Thank you for your question regarding the key advancements of our study. Our work demonstrates several significant breakthroughs compared to previous MutaT7 systems:

(1) Comprehensive deaminase optimization

We systematically tested cytosine (PmCDA1-UGI⁹, evoPmCDA1-UGI¹⁰) and adenine (TadA7.10¹¹, TadA8e¹², and TadA9¹³) deaminases, as well as dual-type deaminases (CABE T3.1¹⁴, T3.155¹⁴, TadDE¹⁵) in the non-model organism *H. bluephagenesis*. The top performers (PmCDA1-UGI and TadA8e) were combined with orthogonal RNA polymerases to create mutators yielding all transition mutations efficiently.

(2) Higher mutation efficiency

Our optimized system (pMT23opt-MmP1) achieves 2.64 transition mutations/day/kb, outperforming MutaT7^{transition5} (1.58 mutations/day/kb), dual8⁶ mutator (0.46 mutations/day/kb), and MutaT7^{GDE6} (1.28 mutations/day/kb). This enhanced efficiency enables rapid exploration of protein fitness landscapes, accelerating the discovery of functional variants.

(3) Broader applicability

Unlike MutaT7, our orthogonal transcription mutators function effectively in both *E. coli* and the non-model organism *H. bluephagenesis*, expanding their utility in synthetic biology and metabolic engineering. To our knowledge, this is the first report of an efficient targeted evolution system developed for extremophiles. Additionally,

the strong orthogonality of our three-phage RNA polymerase-based system further supports modular design for diverse applications.

(4) Rapid mutant generation

Due to the high mutagenesis efficiency, a single 24-hour induction is sufficient to generate rich mutant libraries, eliminating the need for continuous passaging (as in MutaT7). We validated this with fluorescent proteins, chromoproteins, cytoskeletal proteins, cell division-related proteins, the sigma 70 factor, and the arginine exporter, demonstrating visible efficiencies.

Collectively, our system offers optimized deaminase combinations, higher efficiency, broader host compatibility, and faster mutant generation—particularly valuable for non-model organisms.

5) Please discuss prior methods (e.g., CRISPR-based mutagenesis, error-prone PCR, etc.) in comparison to your results.

We sincerely appreciate your good question regarding the comparison of our method relative to existing mutagenesis techniques. We have now included the discussion of these comparisons in the revised manuscript (Pages 23-26, Lines 440-447, 481-484, and 498-500), and would like to summarize the key points here:

(1) Compared to CRISPR-based methods (e.g., EvolvR¹⁶):

Our system achieves mutagenesis across longer target sequences (e.g., 1.4 kb *sacB* gene and 2.2 kb fluorescent protein genes) than the typical 350 nt editing window of EvolvR¹⁶.

(2) Compared to error-prone *PCR*¹⁷:

Our *in vivo* approach eliminates the need for time-consuming library construction and transformation, overcoming the bottleneck of low transformation efficiency that limits library diversity in error-prone *PCR* methods.

(3) Compared to the MutaT7 system^{5, 6}:

Our system demonstrates higher mutation efficiency and broader host compatibility, particularly in non-model organisms like *H. bluephagenesis*.

6) Finally, can this toolkit be applied to a eukaryotic system?

Thank you for raising this important question regarding potential eukaryotic applications of our toolkit. While our current study focused on prokaryotic systems (*E. coli* and *H. bluephagenesis*), we believe extension to eukaryotic systems represents a promising direction for future research. The successful adaptation of the MutaT7 system in eukaryotic cells suggests that our orthogonal transcription-based system may similarly function within eukaryotic cells, given the evolutionary conservation of these phage RNA polymerases¹⁸.

References:

1. Ravikumar, A., Arzumanyan, G.A., Obadi, M.K., Javanpour, A.A. & Liu, C.C. Scalable, continuous evolution of genes at mutation rates above genomic error thresholds. *Cell* **175**, 1946-1957. e1913 (2018).
2. Tian, R. et al. Establishing a synthetic orthogonal replication system enables accelerated evolution in *E. coli*. *Science* **383**, 421-426 (2024).
3. Sarkar, S., Ma, W.T. & Sandri, G.v.H. On fluctuation analysis: a new, simple and efficient method for computing the expected number of mutants. *Genetica* **85**, 173-179 (1992).
4. Radchenko, E.A., McGinty, R.J., Aksenova, A.Y., Neil, A.J. & Mirkin, S.M. in *Genome Instability: Methods and Protocols*. (eds. M. Muzi-Falconi & G.W. Brown) 421-438 (Springer New York, New York, NY; 2018).
5. Seo, D., Koh, B., Eom, G.-e., Kim, H.W. & Kim, S. A dual gene-specific mutator system installs all transition mutations at similar frequencies in vivo. *Nucleic Acids Research* **51**, e59-e59 (2023).
6. Mengiste, A.A. et al. MutaT7GDE: A Single Chimera for the Targeted, Balanced, Efficient, and Processive Installation of All Possible Transition Mutations In Vivo. *ACS Synthetic Biology* **3**, 2693-2701 (2024).
7. Alper, H., Moxley, J., Nevoigt, E., Fink, G.R. & Stephanopoulos, G. Engineering Yeast Transcription Machinery for Improved Ethanol Tolerance and Production. *Science* **314**, 1565-1568 (2006).
8. Alper, H. & Stephanopoulos, G. Global transcription machinery engineering: A new approach for improving cellular phenotype. *Metabolic Engineering* **9**, 258-267 (2007).
9. Rogozin, I.B. et al. Evolution and diversification of lamprey antigen receptors: evidence for involvement of an AID-APOBEC family cytosine deaminase. *Nature Immunology* **8**, 647-656 (2007).
10. Thuronyi, B.W. et al. Continuous evolution of base editors with expanded target compatibility and improved activity. *Nature Biotechnology* **37**, 1070-1079 (2019).
11. Gaudelli, N.M. et al. Programmable base editing of A•T to G•C in genomic DNA without DNA cleavage. *Nature* **551**, 464-471 (2017).
12. Richter, M.F. et al. Phage-assisted evolution of an adenine base editor with improved Cas domain compatibility and activity. *Nature Biotechnology* **38**, 883-891 (2020).
13. Yan, D. et al. High-efficiency and multiplex adenine base editing in plants using new TadA variants. *Molecular Plant* **14**, 722-731 (2021).
14. Neugebauer, M.E. et al. Evolution of an adenine base editor into a small, efficient cytosine base editor with low off-target activity. *Nature biotechnology* **41**, 673-685 (2023).
15. Lam, D.K. et al. Improved cytosine base editors generated from TadA variants. *Nature Biotechnology* **41**, 686-697 (2023).
16. Halperin, S.O. et al. CRISPR-guided DNA polymerases enable diversification of all nucleotides in a tunable window. *Nature* **560**, 248-252 (2018).

17. Cirino, P.C., Mayer, K.M. & Umeno, D. Generating mutant libraries using error-prone PCR. *Directed evolution library creation: Methods and protocols* **231**, 3-9 (2003).
18. Chen, H. et al. Efficient, continuous mutagenesis in human cells using a pseudo-random DNA editor. *Nature Biotechnology* **38**, 165-168 (2020).

Response to Reviewer Comments

We are grateful to the reviewers for their constructive feedback and valuable recommendations, which have significantly enhanced our manuscript. Below we provide detailed responses to each of their comments.

Reviewer #1 (Remarks to the Author):

The reviewer's concerns have been properly addressed.

We sincerely appreciate the reviewer's valuable time and insightful comments on our manuscript.

Reviewer #2 (Remarks to the Author):

The authors have addressed all previous comments, and the revised manuscript shows clear improvement in clarity and quality. I find the revisions appropriate and consider the manuscript suitable for publication.

We sincerely appreciate the reviewer's positive evaluation of our revisions and kind recommendation for publication. We are grateful for the time and expertise they have contributed to improving our manuscript.

Reviewer #3 (Remarks to the Author):

Accepted after revision.

We sincerely appreciate the opportunity to revise our manuscript and are grateful for the constructive feedback that helped improve our work.